# A New Edge Betweenness Measure Using a Game Theoretical Approach: An Application to Hierarchical Community Detection

**Daniel Gómez** [1,2] , **Javier Castro** [1,2] , **Inmaculada Gutiérrez** [1,*] **and Rosa Espínola** [1,2]

1 Faculty of Statistics, Complutense University Puerta de Hierro, 28040 Madrid, Spain; dagomez@estad.ucm.es (D.G.); jcastroc@estad.ucm.es (J.C.); rosaev@estad.ucm.es (R.E.)
2 Instituto de Evaluación Sanitaria, Complutense University, 28040 Madrid, Spain
\* Correspondence: inmaguti@ucm.es

**Abstract:** In this paper we formally define the *hierarchical clustering network problem* (HCNP) as the problem to find a *good* hierarchical partition of a network. This new problem focuses on the dynamic process of the clustering rather than on the final *picture* of the clustering process. To address it, we introduce a new hierarchical clustering algorithm in networks, based on a new shortest path betweenness measure. To calculate it, the communication between each pair of nodes is weighed by the importance of the nodes that establish this communication. The weights or importance associated to each pair of nodes are calculated as the Shapley value of a game, named as *the linear modularity game.* This new measure, (*the node-game shortest path betweenness measure*), is used to obtain a hierarchical partition of the network by eliminating the link with the highest value. To evaluate the performance of our algorithm, we introduce several criteria that allow us to compare different dendrograms of a network from two point of view: modularity and homogeneity. Finally, we propose a faster algorithm based on a simplification of *the node-game shortest path betweenness measure*, whose order is quadratic on sparse networks. This fast version is competitive from a computational point of view with other hierarchical fast algorithms, and, in general, it provides better results.

**Keywords:** game theory; graph theory; hierarchical clustering networks; community detection problems; divisive algorithms

## 1. Introduction

A major issue in the field of Social Network Analysis (SNA) is the search and study of cohesive groups in a network [1]. This issue has been extended to networks of different nature, including biological, communication, transportation and social networks, among many others. Several authors have addressed this problem using a community structure detection approach [1–3]. Communities, also known as clusters or modules, are groups of vertices or nodes that probably share common properties and/or play similar roles within the graph.

Clustering network algorithms can be divided into non-hierarchical (usually referred to as partitional clustering) and hierarchical methods which consider the final output. The main difference between these types of algorithms is the result obtained. Partitional clustering methods focus on the final partition that divides the set of items into homogeneous groups. Hierarchical clustering methods present an evolution (by means of a dendrogram) of how the groups are formed (in agglomerative methods) or split (in divisive methods), from the beginning of the process to the final step. Although hierarchical clustering methods present more computational problems, they have an important advantage, as the obtained results are more informative than those given by partitional clustering algorithms, which only provide a final picture of the process. Thus, in this paper we focus on techniques for hierarchical clustering of a set of nodes in a network as well as the mea-

sures that facilitate that hierarchical clustering. Initially, the methods proposed to obtain hierarchical partitions of communities in networks by clustering processes, considered the calculation of the weight $w_{ij}$ for all the pairs $(i, j)$, being $i, j$ two nodes [1,4]. When all these weights are calculated, and once this dissimilarity or distance $W$ matrix is built, the vertices can be grouped by means of any classical hierarchical clustering technique, without considering the structure of the network. For example, let $G = (N, E)$ denote a graph, so that $N$ denotes the set of nodes and $E$ denotes the set of edges. Therefore, a good way to hierarchically cluster the set of nodes $N$ is by starting with that pair of nodes $i, j$ whose corresponding weight $w_{ij}$ is the strongest of all, and then go progressing to the weakest a review of classical hierarchical techniques can be found in [5]. The output is a hierarchical cluster that can be visualized in a dendrogram.

Those preliminary techniques offered a wide range of possibilities. Nevertheless, it is important to bear in mind the drawbacks that result from a full consideration of the structure. In [2], Newman and Girvan provide a description of the problems derived from those techniques, as for example the memory required, $\binom{n}{2}$ instead of the amount of edges in $E$, and several assumptions on which the characterization of matrix $M$ is based.

Then, Girvan and Newman proposed a divisive algorithm [4] (GN on the following) with a good performance for small-medium networks. It is based on the calculation of a weight $w_{ij}$ for each pair of adjacent nodes in the network (nodes $i, j \in N$ are adjacent if $\exists \{i, j\} \in E$), and not for all node pairs, as traditional methods do. The weight $w_{ij}$ assigned to the corresponding edge $\{i, j\}$ shows its power in the communication structure determined by the graph. So, firstly the weight $w_{ij}$ has to be calculated for all the edges $\{i, j\} \in E$, i.e., for all the pairs of adjacent nodes $i, j$. Hence, the divisive algorithm can be summarized as follows:

1. For every edge $\{i, j\} \in E$, calculate the weight $w_{ij}$.
2. Remove the edge $\{k, \ell\}$, being $w_{k,\ell} = \max\{w_{ij} \mid \{i, j\} \in E\}$
3. Calculate the weight $w_{ij}$ for every edge affected by the deleted of $\{k, \ell\}$.
4. Go again to the step 2 and repeat the process until there are no more edges.

The method previously illustrated provides a hierarchical clustering of the nodes of the network. An essential notion when calculating $w_{ij}$ is the betweenness, described in [4] as the frequency with which every edge takes action in the communication of the set of nodes. There are two ways to obtain that frequency: the shortest path betweenness (SP), which only considers geodesic parts; or the random-walk betweenness defined in [6]. These options are some of the most important when calculating the weight $w_{ij}$; however, different options which procure several modifications of the divisive algorithm have been proposed in the literature [7–9].

Let us note that most of the methods used to calculate a weight or measure $w_{ij}$ for each link in hierarchical clustering assume that all the communications between any pair of nodes in the network are equally important [2,4,6–9]. In our opinion, it is obvious that this assumption is unrealistic for networks that represent real-world communication situations.

To avoid this assumption, our main contribution is to estimate the relevance or importance of the communications between any pair of nodes in a graph, focusing on situations in which the only information available is a graph. We consider that communication among *important* nodes will be more relevant than communication among *dummy* nodes. To determine the importance of a node in a given network, we use centrality or power measures (see [10,11] for a general review of centrality measures). When it comes to problems modeled by a network, it is very common to use some tools inherited from game theory [12,13]. In particular, cooperative game theory has been used as natural approach to represent the power or centrality of nodes in a network (see for example [14,15]), or also the cohesion of a set of nodes [16]. The use of Game Theory for solving community detection problems is not new. For example, ref. [17] Avrachenkov et al. proposed two approaches based on the Myerson value of two games: one of them based on simple paths and the other based on hedonic games. This Myerson value permits the authors to define a concept

of Nash stable partition which provides a solution to the clustering problem. Once the differences in communications are determined, we obtain a new betweenness measure for each link by incorporating in the classical concept the notion that communications between any pair of nodes are not equally weighted.

The remainder of the paper is organized as follows. In Section 2 we introduce some preliminaries concepts in community detection problems and game theory, introducing a formal characterization of the *Hierarchical Clustering Network Problem*. In Section 3 we define a new SP betweenness measure based on game theory, which is used to obtain a new hierarchical clustering algorithm. In Section 4, we analyze the computational complexity associated to this new SP betweenness measure as well as the divisive algorithm based on it. In Section 5 we define a faster version of the algorithm previously proposed in Section 3, analyzing also its computational complexity. In Section 6, we compare the divisive algorithm obtained from the new SP betweenness measure with some of the most well-known hierarchical clustering algorithms in some well-known network examples. We also propose several criteria to compare the results given by the different hierarchical algorithms focusing, on the modularity and the homogeneity of the partitions. In Section 7 we analyze the performance of the algorithms here presented in the GN benchmark. We discuss the results of the paper in Section 8, and then we finish in Section 9 with some conclusions.

## 2. Preliminaries

### 2.1. Cooperative Game Theory

Let $N = \{1, 2, \ldots, n\}$ denote a finite set of players. A game in characteristic function form (a coalitional game or a TU game) is a pair $(N, v)$ where $v$ a real function known as characteristic function and defined on $2^N$, the set of all subsets of $N$ (coalitions), that satisfies $v(\varnothing) = 0$. For each $S \subseteq 2^N$, $v(S)$ represents the economic possibilities of players in $S$. $G^N$ will denote the class of all coalitional games with players set $N$. When there is no ambiguity regarding $N$, we will refer to the game $(N, v)$ as $v$.

Probably, the most important solution concept in cooperative games was defined by Shapley in [18]. This solution concept is known as the Shapley value. The Shapley value is useful when there exists a need to allocate the worth that a set of players can achieve if they agree to cooperate. The Shapley value can be used in a natural way to measure the power or the importance of individuals.

**Definition 1** (Shapley value [18]). *Let $N$ denote a finite set of players, and let the TU game $(N, v)$. The Shapley value of the player $i \in N$ is defined as*

$$\sum_{S \subset N\, i \notin S} Sh_i(v) = \frac{|S|!(|N| - 1 - |S|)!}{|N|!} [v(S \cup \{i\}) - v(S)] \tag{1}$$

Other solutions concepts have been proposed to represent the power, importance or capacity of individuals in a cooperative game. Based on the similar structure of marginal contributions of previous definition, in [19] it was defined the concept of semivalue.

**Definition 2** (Semivalue [19]). *Let $(N, v)$ denote a cooperative game, and let $i \in N$ denote a player. The semivalue $\Psi_i(v)$ is defined as:*

$$\sum_{S \subset N \setminus \{i\}} P(S)[v(S \cup \{i\}) - v(S)] \tag{2}$$

*where $\displaystyle\sum_{S \subset N \setminus \{i\}} P(S) = 1$ and $P(S) \geq 0, \forall S \subset N \setminus \{i\}$*

Let us observe that $\Psi_i(v)$ is a convex combinations of the marginal contributions of the player $i$ regarding the different coalitions $S \subset N \setminus \{i\}$, where the values of this convex

combination only depends on the cardinality of $S$. Obviously, the Shapley value is a specific case of a semivalue but it is not the only solution concept that derives from a semivalue. The *Banzhaf-Coleman* value [20] is another well-known semivalue which has been deeply studied in the literature [21]. Formally, the Banzhaf-Coleman value is a semivalue where:

$$P(S) = \frac{1}{2^n} \quad \forall S \subset N \setminus \{i\}. \tag{3}$$

*2.2. Hierarchical Clustering for Graphs*

A graph or a network $G$ is a pair $(N, E)$, where $N = \{1, 2, \ldots, n\}$ denotes a finite set of nodes and $E$ is a collection of links or edges, that is, unordered pairs $\{i, j\}$ with $i, j \in N$. The adjacency matrix of a graph, usually denoted by $A$, is another way to characterize a graph. This matrix represents the direct connections among nodes, so that $A_{ij} = 1$ if there exist the edge $\{i, j\} \in E$ and 0 otherwise.

Given a graph or network, the problem of finding a *good* partition on it is known as clustering network problem or community detection problem. Although it can be found other definitions of this problem in which nodes may belong to more than one partition, in the classical approach only non overlapping partitions are allowed, so a feasible partition is as follows.

**Definition 3** (Feasible partition). *A partition P of a graph $(N, E)$ is defined as a set $P = \{H_1, \ldots, H_r\}$, where for all $i \neq j$, $H_i \cap H_j = \varnothing$ (non overlapping communities), $\bigcup_{j=1}^{r} H_j = N$, and for all i, the subgraph $(H_i, E_{|H_i})$ is a connected graph (i.e., there exist, at least, one path between any pair of nodes of $H_i$ that does not use any node outside of $H_i$ ).*

In this paper, we focus on the Hierarchical Clustering Network Problem (HCNP). It can be defined as the problem of finding a *good* hierarchical clustering of the graph, it is, a sequence of partitions build in a hierarchical way. A hierarchical partition of a graph is usually represented by a dendrogram. Then we introduce several concepts and notation, for a next formal definition of the hierarchical clustering of a graph.

**Definition 4** (Finer partition). *Let P and Q denote two partitions of a graph. P is said to be finer than Q if for all $A \in P$, there exists $B \in Q$ so that $A \subseteq B$. If P is finer than Q, we denote it by $P\widetilde{\subseteq}Q$.*

**Definition 5** (Hierarchical partition). *Let $G = (N, E)$ denote a graph with k connected components, and let $\mathcal{D} = (P_1, P_2, \ldots, P_{n-k})$ denote a sequence of partitions of G. $\mathcal{D}$ is said to be a hierarchical partition of the graph (obtained in a divisive way) when the following holds:*

- *$P_{n-k} = N$ (i.e., in the last partition, all the nodes are singleton clusters).*
- *For all $i = 1, \ldots, n - k - 1$, $|P_{i+1}| = |P_i| + 1$. (i.e., the number of groups or communities increases in one unit in each iteration).*
- *$P_{i+1}\widetilde{\subseteq}P_i$ for every $i = 1, \ldots, n - k - 1$.*

Another way to define this hierarchical clustering of a network is by changing the arrangement of the sequence of partitions. Note that this options is also an agglomerative process. However, it is obvious that every hierarchical clustering network algorithm, whether agglomerative or divisive, provides a hierarchical partition of the graph in this way. The set of all the possible hierarchical clustering for a given network $(N, E)$ is denoted by $\mathcal{P}(\mathcal{D})$.

**Example 1.** *Let $G = (N, E)$ denote a graph with $N = \{1, 2, 3, 4\}$ and $E = \{\{1, 2\}, \{1, 3\}, \{2, 4\}, \{3, 4\}\}$, as showed in the Figure 1.*

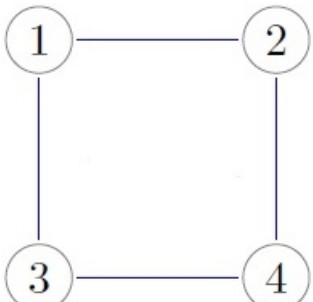

**Figure 1.** Graph $G = (N, E)$.

*A possible hierarchical partition of this graph is:*

- $\mathcal{D}_1 = \{\mathcal{P}^0, \mathcal{P}^1, \mathcal{P}^2, \mathcal{P}^3\}$ *with:*

$\mathcal{P}^0 = \Big\{\{1,2,3,4\}\Big\}$
$\mathcal{P}^1 = \Big\{\{1\}, \{2,3,4\}\Big\}$
$\mathcal{P}^2 = \Big\{\{1\}, \{2,4\}, \{3\}\Big\}$
$\mathcal{P}^3 = \Big\{\{1\}, \{2\}, \{3\}, \{4\}\Big\}$

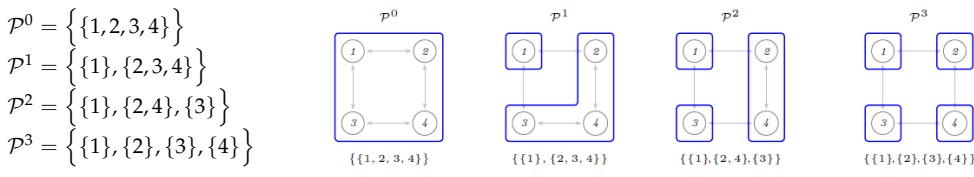

*The following sequences of partitions are not hierarchical partitions of G.*

- $\mathcal{D}_2 = \{\mathcal{S}^0, \mathcal{S}^1, \mathcal{S}^2, \mathcal{S}^3\}$ *with:*

$\mathcal{S}^0 = \Big\{\{1,2,3,4\}\Big\}$
$\mathcal{S}^1 = \Big\{\{1,3\}, \{2,4\}\Big\}$
$\mathcal{S}^2 = \Big\{\{1\}, \{2,3\}, \{4\}\Big\}$
$\mathcal{S}^3 = \Big\{\{1\}, \{2\}, \{3\}, \{4\}\Big\}$

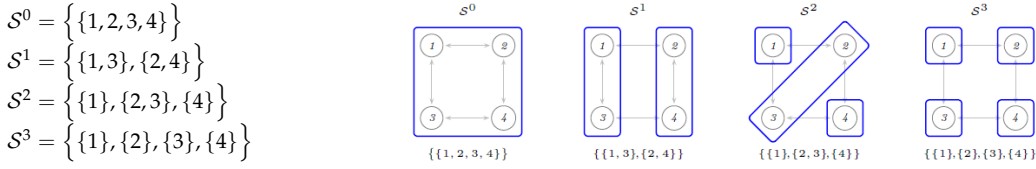

*Note that* $\{2\}$ *and* $\{3\}$ *are not connected in G and, consequently, it is not a partition.*

- $\mathcal{D}_3 = \{\mathcal{T}^0, \mathcal{T}^1, \mathcal{T}^2, \mathcal{T}^3\}$ *with:*

$\mathcal{T}^0 = \Big\{\{1,2,3,4\}\Big\}$
$\mathcal{T}^1 = \Big\{\{1,2\}, \{3,4\}\Big\}$
$\mathcal{T}^2 = \Big\{\{1,3\}, \{2\}, \{4\}\Big\}$
$\mathcal{T}^3 = \Big\{\{1\}, \{2\}, \{3\}, \{4\}\Big\}$

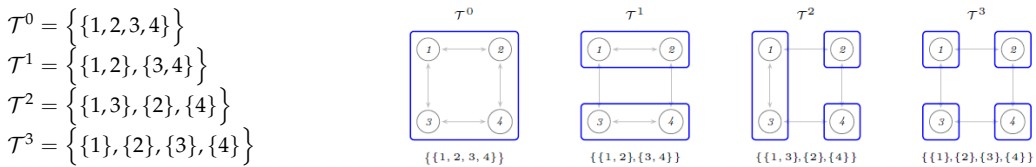

*Note that* $\mathcal{T}^2 \nsubseteq \mathcal{T}^1$, *so it is not a hierarchical partition.*

Once hierarchial clustering network problem has been clearly defined, we review different clustering networks algorithms, focusing on those that produce hierarchical partitions. Some of the most well-known algorithms in community detection can be found in [1,22]. As it is pointed in [22], these algorithms are a representative subset, that exploits some of the most interesting ideas and techniques developed over the last years.

1. Girvan and Newman algorithm (see introduction for more details), denoted as *GN* algorithm in the following.
2. Clauset, Newman and Moore algorithm (see [23] for more details), denoted as *CNM* algorithm in the following.
3. Walktrap algorithm. Based on random walk (see [24] for more details), denoted as *Walktrap* algorithm in the following.

4.　Newman 2012 algorithm (see [25] for more details), denoted as *N*2012 algorithm in the following. This algorithm is considered as a reference of those algorithms that are based on the construction of a dissimilarity matrix between any pairs of nodes.
5.　Radicchi et al. algorithm (see [8] for more details), denoted as *Radicchi* algorithm in the following. In [8], two measures to cut the edges of the graph giving the weak and strong version of the *Radicchi* algorithm are proposed. For the computational results of this paper, we have used (in each example) the version that obtain better results.

Clearly, the methods *GN*, *CNM*, *Walktrap*, *N*2012 and *Radicchi* can be understood as hierarchical clustering methods, as a hierarchical partition of the network is obtained with all of them, according to the basis of the Definition 5. There are other methods which have not been considered in the current work because they do not give a hierarchical partition in the sense of Definition 5: some of them (as for example the well-known *Louvain algorithm* [26] or the *Leiden method* [27]) do not provide a hierarchical partition in which the amount of communities increases in one unit in each iteration. On the other hand, there are other type of methods,a s those based on eigenvector transformation [28], which do not guarantee connected partitions in every step. Lately, some authors have addressed the problem in which the network has not a clear community structure, and many methods have been proposed; however, this approach is not comparable to ours [29,30].

### *2.3. Modularity Function in Networks*

The measurement of the reliability of a partition is not easy; moreover, there is no a unique global measure. Here we focus on the definition of the modularity (usually denote as $Q$) reported in [2]. The modularity is very often used to measure partition goodness. Then, given a partition of a network, $P$, the corresponding modularity is defined as:

$$Q_P = \frac{1}{2m} \sum_{i,j} \left[ A(i,j) - \frac{l_i \, l_j}{2m} \right] \delta(c_i \, c_j) \tag{4}$$

where $m$ denotes the number of edges in the graph, $l_i$ denotes the degree of the node $i$, $A$ represents the adjacency matrix of the graph, and the value of $\delta(c_i, \, c_j)$ depends on if $i$ and $j$ belong to the same cluster, in the sense that $\delta(c_i, \, c_j) = 1$ if $i$ and $j$ are in the same community, and $\delta(c_i, \, c_j) = 0$ otherwise. On the other hand, the modularity of a partitions measures the difference between the amount of edges which are in the considered groups, and the expected amount of edges which would be in those groups if the graph were randomly distributed.

Taking into account that hierarchical divisive methods provide a partition in each step, there is a modularity vector related to a hierarchical clustering; it is denoted by Q. The coordinate $\ell$ of this vector represent the modularity of the partition obtained in this step.

Formally, given a hierarchical clustering of a network $\mathcal{D} = \{P_1, \dots, P_{n-k}\}$, the modularity vector can be defined as $\mathcal{Q} = (Q_1, \dots, Q_{n-k})$ where $Q_i$ is the modularity of the partition $P_i$.

**Example 2.** *Let $G = (N, E)$ denote a graph where $N = \{1, 2, 3, 4\}$ and $E = \{\{1, 2\}, \{1, 3\}, \{3, 4\}, \{2, 4\}\}$ and let $\mathcal{D} = \{P_1, P_2, P_3\}$ with $P_1 = \{\{1\}, \{2, 3, 4\}\}$, $P_2 = \{\{1\}, \{2, 4\}, \{3\}\}$, $P_3 = \{\{1\}, \{2\}, \{3\}, \{4\}\}$ a hierarchical partition of the graph $(N, E)$. Then modularity vector $\mathcal{Q}(D)$ associated to $\mathcal{D}$ is $\mathcal{Q}(D) = (Q_1, Q_2, Q_3) = (Q(P_1), Q(P_2), Q(P_3))$ which is equal to $(-0.125, -0.125, -0.25)$:*

### *2.4. The SP Edge Betweenness Measure*

The SP betweenness measure defined in [4] is an extension of the popular concept of site betweenness introduced in [31]. SP edge betweenness represents the importance of the edges in processes such as information spread, where information usually flows through SPs. Thus, an edge or link has a high SP betweenness value if it lies on a large number of short paths between vertices.

**Definition 6** (SP betweenness measure [4])**.** *Let $G = (N, E)$ denote a graph, and let the pair of adjacent nodes $i, j$. The SP betweenness measure of the link $\{i, j\}$ is defined as:*

$$w_{ij}^B = \sum_{\substack{r,s \in N \\ r \neq s}} I_{rs}((ij)), \qquad (5)$$

*where $I_{rs}((ij)) = 1$ if the SP from $r$ to $s$ uses the link $\{i, j\}$. If there are more than one SPs between $r$ and $s$, $I_{rs}((ij))$ is divided by the number of SPs.*

Although the SP betweenness measure performs well in the *GN* algorithm, there are some real situations in which it yields poor results in terms of modularity. In general, such poor performance occurs when there are nodes of *low* degree (usually 1) connected to a set of nodes for which the edges are redundant. In these situations, the SP betweenness is greater for the link that connects an isolated node with the cluster than for the rest of the links or for connections within the cluster. Thus, after applying the *GN* algorithm with this measure, nodes of low degree are cut from the graph. This yields two clusters with cardinality 1 and $n - 1$, where $n$ is the number of nodes in the graph. This situation is now illustrated with an example.

**Example 3.** *Let $G = (N, E)$ denote a graph with $N = \{1, 2, 3, 4, 5, 6\}$ $E = \{(1, 2), (2, 3), (2, 4), (3, 5), (4, 5), (5, 6)\}$. The SP betweenness measure for the links of graph $G = (N, E)$ reaches a maximum value for the edges $(1, 2)$ and $(5, 6)$. Thus, according to the GN algorithm, one of the edges $(1, 2)$ or $(5, 6)$ is removed randomly in the first step. It is easy to see that this first partition has low quality in terms of modularity ($Q = \frac{-1}{72}$). In the second step, after the recalculation of the betweenness measure for the links in the new graph, the GN algorithm cuts the other link, dividing the graph into three components, $\{1\}$, $\{6\}$ and $\{2, 3, 4, 5\}$ with modularity $Q = \frac{-3}{72}$. In the following step it is divided the square 2,3,4,5 in two components, for example $\{2, 3\}$ and $\{4, 5\}$ with modularity $Q = \frac{-2}{72}$.*

This poor performance is reproduced in higher-dimensionality problems and complex networks that contain nodes of low degree connected to groups of nodes for which the relations are redundant. In these situations, the relations that connect nodes of low degree are considered more important (in terms of betweenness measure) than others, so the divisive algorithm defined in [2] starts cutting these nodes from the rest of the network, leading to poor partition. Although this could be reasonable from a betweenness measure point of view, it is clear that the obtained partition is poor in terms of modularity and size homogeneity when this measure is used as a criterion to cut the graph.

## 3. A New SP Edge Betweenness Measure Based on Game Theory

When calculating the SP edge betweenness measure, all communications are considered equally important. However, for many real situations this assumption is far from reality [32]. It is easy to imagine real networks (transportation networks, flight communications networks and social networks, among others) for which communications between key nodes are more relevant than others.

We denote the importance of the communication between nodes $r$ and $s$ (not necessarily adjacent nodes) by $h_{rs}$. We emphasize that if the only information available for a real network is the direct relations between its nodes, then classical methods [2,4,6–9] assume that $h_{rs} = 1$ for all $r \neq s$ to obtain a weight $w_{ij}$ for each link of the graph. This is one of the main reasons for the poor performance of the SP betweenness measure noted above.

To estimate the value of $h_{rs}$ for each pair of nodes $r$ and $s$ with any additional information, we assume that the communications between main nodes/actors (paths among them) are more relevant than communications between dummy nodes or between important nodes and dummy nodes. Thus, before determining the $h_{rs}$ value for a pair of nodes $r$ and $s$, it is necessary to measure the importance of the nodes in the graph. To this

aim, we use centrality measures to represent the importance of nodes in a network. Let $\Phi = (\phi_1, \ldots, \phi_n)$, denote the vector that measures the node importance in a graph.

As previously pointed out, it seems logical that the relations among powerful nodes should be considered more relevant than others. Thus, to aggregate the values $\phi_r$ and $\phi_s$ to determine the importance of the relation $h_{rs}$, we use conjunctive aggregation operators. Although many aggregation operators can be used [33], we apply the most well known, the minimum operator. Then the importance of the communication between nodes $r$ and $s$ is calculated as $h_{rs} = \ MIN \ \{\phi_r, \phi_s\}$.

Once the weight $h_{rs}$ has been determined for all pairs of nodes in the network, the new SP edge betweenness measure that we will refer as *node-weighted SP edge betweenness* measure can be expressed as:

$$w_{ij} = \sum_{(r,s)r\neq s} h_{rs} I_{rs}((ij)),$$

where $I_{rs}((ij)) = 1$ if the SP from $r$ to $s$ uses the link $\{i, j\}$. Note that this formula is the same that formula (5) if $h_{rs} = 1$. It is important to note that, as occurs in the original Girvan–Newman SP betweenness measure, if there is more than one SP between nodes $r$ and $s$, $I_{rs}((ij))$ is divided by the number of SPs.

*Determining the Importance of the Nodes: A Game Theoretical Approach*

Cooperative game theory can be a useful tool in order to represent the importance of nodes in a network (see for example [34] among other approaches), when it is known the motivation (the game) that shows the interaction between players. Different games may be proposed in this section, but, taking into account that our main aim is to maximize the modularity of a partition, we will focus on those games that show the power or importance of a coalition in the modularity function and/or the community detection problem.

First of all, let us observed that modularity formula defined in (4) is equivalent to

$$Q_P = \frac{1}{2m} \sum_{r=1}^{k} \sum_{i,j\in C_r} \left[ A(i,j) - \frac{l_i\, l_j}{2m} \right] \tag{6}$$

Furthermore, as it is well known in the literature [35,36] and demonstrated in the Appendix A, note that Equation (6) is equivalent to

$$Q_P = \sum_{r=1}^{k} \left[ \frac{l_{in}(C_r)}{m} - \left[ \frac{l_{in}(C_r)}{m} + \frac{l_{out}(C_r)}{2m} \right]^2 \right] \tag{7}$$

where $l_{in}(C_r)$ denotes the number of edges of the set of nodes inside the cluster $C_r$ (i.e., $l_{in}(C_r) = \sum_{i,j\in C_r} A(i,j)/2$) and $l_{out}(C_r)$ denotes the number of edges that go from the cluster $C_r$ to outside (i.e., $l_{out}(C_r) = \sum_{i\in C_r, j\notin C_r} A(i,j)$).

From Equation (7), it is clear that the contribution of a coalition $C_r$ to the modularity function only depends on two values: $l_{in}(C_r)$ and $l_{out}(C_r)$, so the corresponding game that shows the importance or power of a coalition should be strongly dependent on these two parameters.

We provide the following example to illustrate the definition of the modularity, according to Equations (6) and (7).

**Example 4.** *Let $G = (N, E)$ denote the graph showed in the Figure 2, and let the partition $P = \{\{1,2,3,4\}, \{5,6,7\}\}$. Here we provide the calculation of the modularity of the partition $P$ by considering the formulas in Equations (6) and (7).*

$$A = \begin{pmatrix} 0 & 1 & 1 & 1 & 0 & 0 & 0 \\ 1 & 0 & 1 & 1 & 0 & 0 & 0 \\ 1 & 1 & 0 & 1 & 0 & 0 & 0 \\ 1 & 1 & 1 & 0 & 1 & 0 & 0 \\ 0 & 0 & 0 & 1 & 0 & 1 & 1 \\ 0 & 0 & 0 & 0 & 1 & 0 & 0 \\ 0 & 0 & 0 & 0 & 1 & 1 & 0 \end{pmatrix}$$

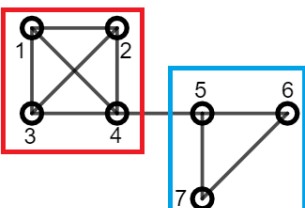

**Figure 2.** Graph $G = (N, E)$ with 7 nodes, and the considered partition, $P = \{\{1,2,3,4\}, \{5,6,7\}\}$.

▷ *Using Equation* (6): $Q_P = \frac{1}{2m} \sum_{r=1}^{k} \sum_{i,j \in C_r} \left[ A(i,j) - \frac{l_i l_j}{2m} \right] = \frac{1}{2*10} \left( \left( \overbrace{0 - \frac{3*3}{2*10}}^{1,1 \in C_1} \right) + \right.$

$$\left( \overbrace{1 - \frac{3*3}{2*10}}^{1,2 \in C_1} \right) + \left( \overbrace{1 - \frac{3*3}{2*10}}^{1,3 \in C_1} \right) + \left( \overbrace{1 - \frac{3*4}{2*10}}^{1,4 \in C_1} \right) + \left( \overbrace{1 - \frac{3*3}{2*10}}^{2,1 \in C_1} \right) + \left( \overbrace{0 - \frac{3*3}{2*10}}^{2,2 \in C_1} \right) +$$

$$\left( \overbrace{1 - \frac{3*3}{2*10}}^{2,3 \in C_1} \right) + \left( \overbrace{1 - \frac{3*4}{2*10}}^{2,4 \in C_1} \right) + \left( \overbrace{1 - \frac{3*3}{2*10}}^{3,1 \in C_1} \right) + \left( \overbrace{1 - \frac{3*3}{2*10}}^{3,2 \in C_1} \right) + \left( \overbrace{0 - \frac{3*3}{2*10}}^{3,3 \in C_1} \right) +$$

$$\left( \overbrace{1 - \frac{3*4}{2*10}}^{3,4 \in C_1} \right) + \left( \overbrace{1 - \frac{4*3}{2*10}}^{4,1 \in C_1} \right) + \left( \overbrace{1 - \frac{4*3}{2*10}}^{4,2 \in C_1} \right) + \left( \overbrace{1 - \frac{4*3}{2*10}}^{4,3 \in C_1} \right) + \left( \overbrace{0 - \frac{4*4}{2*10}}^{4,4 \in C_1} \right) +$$

$$\left( \overbrace{0 - \frac{3*3}{2*10}}^{5,5 \in C_2} \right) + \left( \overbrace{1 - \frac{3*2}{2*10}}^{5,6 \in C_2} \right) + \left( \overbrace{1 - \frac{3*2}{2*10}}^{5,7 \in C_2} \right) + \left( \overbrace{1 - \frac{2*3}{2*10}}^{6,5 \in C_2} \right) + \left( \overbrace{0 - \frac{2*2}{2*10}}^{6,6 \in C_2} \right) +$$

$$\left. \left( \overbrace{1 - \frac{2*2}{2*10}}^{6,7 \in C_2} \right) + \left( \overbrace{1 - \frac{3*2}{2*10}}^{7,5 \in C_2} \right) + \left( \overbrace{1 - \frac{2*2}{2*10}}^{7,6 \in C_2} \right) + \left( \overbrace{0 - \frac{2*2}{2*10}}^{7,7 \in C_2} \right) \right) = 0.355$$

▷ *Using Equation* (7): $Q_P = \sum_{r=1}^{k} \left[ \frac{l_{in}(C_r)}{m} - \left[ \frac{l_{in}(C_r)}{m} + \frac{l_{out}(C_r)}{2m} \right]^2 \right] =$

$$\overbrace{\left( \frac{6}{10} - \left( \frac{6}{10} + \frac{1}{2*10} \right)^2 \right)}^{C_1} + \overbrace{\left( \frac{3}{10} - \left( \frac{3}{10} + \frac{1}{2*10} \right)^2 \right)}^{C_2} = 0.355$$

One of the main problem that have the most well-known solution concepts in cooperative game theory is the computational complexity related to its calculation. Although this problem was partially solved in [37,38], where the authors proposed some methods to approximate the Shapley value by sampling models in polynomial time, the computational complexity is still high for community detection problems. Then, in order to reduce this computational complexity associated with the calculation of the classical solution concepts in cooperative game theory, we will consider a linear combination of these two values. We will refer to this game as the *linear modularity game*. Taking into account that the modularity function and the community detection problems search partition in which the edges between clusters are punished and the edges inside the cluster are rewarded, we define the linear modularity game as the pair $(N, v_{mod})$, where $N$ is the set of nodes and the characteristic function is given by:

$$v_{mod}(S) = \alpha \frac{l_{in}(S)}{m} - \beta \frac{l_{out}(S)}{m} \quad \forall S \subset N \tag{8}$$

with $\alpha$ and $\beta$ positive values. The characteristic function of the defined linear modularity game models the strength and/or the cohesion of a coalition $S$ as the percentage of links that has this coalition (weighted by $\alpha$) minus the percentage of links that this coalition has with other clusters (weighted by $\beta$). This characteristic function satisfies the two key-properties (see [16] for more details) of any cohesiveness measure: (1) it is increasing in the number of links among its member; (2) it is decreasing in the number of links that connects nodes of the group with others groups.

Let us note that the game $v_{mod}$ is a simplification of the modularity measure; although it has a similar approach, is not the same or equivalent. The similarities between the function $v_{mod}$ and the classic modularity function is that, for a given coalition, both measures punish the relations of its members with the individuals of another coalition and reward the internal relations. However, $v_{mod}$ should clearly be understood as a very simplified version of the original modularity function. We introduce the $v_{mod}$ measure for the sole purpose of simplifying the computational process.

In the following proposition we suggest a specific expression for any semivalue in the linear modularity game that satisfies a small constraint. Firstly, let us introduce some notation. Given a network $(N, A)$, being $A$ the adjacency matrix, and given a node $i \in N$, let us denote by $P_i = \{S \subset N \setminus \{i\}\}$ the set of subsets of $N$ that do not contain the node $i$, and let us denote by $S_i^k = \{R \in P_i \ / \ \sum_{j \in R} A(i, j) = k\}$, the set of subsets in $P_i$ that have exactly $k$ adjacent nodes of $i$. Obviously, the sets $\{S_i^k\}$ establish a partition of $P_i$ for $k = 0, \ldots, l_i$ (i.e $P_i = \bigcup_{k=0}^{l_i} S_i^k$ with $S_i^k \cap S_i^r = \emptyset$ for $k \neq r$). When there is no ambiguity about the node $i$ considered, we denote $S^k = S_i^k$. Before we enunciate the proposition, we provide an example to illustrate the calculation of the values of $S_i^k$.

**Example 5.** *Let $G = (N, E)$ denote the graph showed in the Figure 3*

$$A = \begin{pmatrix} 0 & 1 & 1 & 1 \\ 1 & 0 & 1 & 0 \\ 1 & 1 & 0 & 0 \\ 1 & 0 & 0 & 0 \end{pmatrix}$$

**Figure 3.** Graph $G = (N, E)$ with adjacency matrix $A$. Toy example to illustrate $S_i^k$ calculation.

*We illustrate the calculation of every subset $S_i^k$ with degree $k$ for each single player $i$.*

▷ **Player** $i = 1$

- **k = 0**: $S_1^0 = \{\emptyset\}$
- **k = 1**: $S_1^1 = \{\{2\}, \{3\}, \{4\}\}$
- **k = 2**: $S_1^2 = \{\{2, 3\}, \{2, 4\}, \{3, 4\}\}$
- **k = 3**: $S_1^3 = \{\{2, 3, 4\}\}$

▷ **Player** $i = 2$

- **k = 0**: $S_2^0 = \{\emptyset, \{4\}\}$
- **k = 1**: $S_2^1 = \{\{1\}, \{3\}, \{4\}, \{3, 4\}\}$
- **k = 2**: $S_2^2 = \{\{1, 3\}, \{1, 3, 4\}\}$

▷ **Player** $i = 3$

- **k = 0**: $S_3^0 = \{\emptyset, \{4\}\}$
- **k = 1**: $S_3^1 = \{\{1\}, \{2\}, \{4\}, \{2, 4\}\}$
- **k = 2**: $S_3^2 = \{\{1, 2\}, \{1, 2, 4\}\}$

▷ **Player** $i = 4$

- **k = 0**: $S_4^0 = \{\emptyset, \{2\}, \{3\}, \{2, 3\}\}$

- **k = 1**: $S_4^1 = \{\{1\}, \{1,2\}, \{1,3\}, \{1,2,3\}\}$

Those sets $\{S_i^k\}$ have many properties; the following three are really useful to deal with the Proposition 1 provided below.

- The marginal contribution of player $i$ is the same for every set $\{S_i^k\}$, $\forall k = 0, \ldots, l_i$.
- The number of sets in $\{S_i^k\}$ is equal to the number of sets in $\{S_i^{l_i-k}\}$, $\forall k = 0, \ldots, l_i$.
- The sum of the marginal contribution of an element in $\{S_i^k\}$ and the marginal contribution of an element in $\{S_i^{l_i-k}\}$ turns into an expression which only depends on the degree of the node $i$, i.e., $l_i$.

Then, the proof of the Proposition 1 is based on a sorting of the summation which appears in every semivalue which sets it in terms of $\{S_i^k\}$. After that, these three properties are considered (taking into account if $l_i$ is even or odd) to demonstrate that the summation which appears in the semivalue just depends on $l_i$.

**Proposition 1.** *Let $(N, v_{mod})$ denote the linear modularity game, let $i$ denote a node in $N$ and let $\Psi$ denote a semivalue with $\displaystyle\sum_{S \in S^k} P(S) = \sum_{S \in S^{(l_i-k)}} P(S)$. Then,*

$$\Psi_i(v_{mod}) = \alpha \frac{l_i}{2m}$$

**Proof.** Taking into account that $\Psi_i(v_{mod}) = \displaystyle\sum_{S \subset N \setminus \{i\}} P(S)(v_{mod}(S \cup \{i\}) - v_{mod}(S))$, we first analyze the marginal contribution of the player $i$ in the linear modularity game. We prove that the marginal contribution of a player $i$ in a coalition $S$ is the same for all $S$ that belongs to $S^k$ (i.e., the marginal contribution only depends on the value of $k$).

To do it, we observe that for all $S \in S^k$, $l_{in}(S \cup \{i\}) = l_{in}(S) + k$, and $l_{out}(S \cup \{i\}) = l_{out}(S) - k + (l_i - k)$. So, for all $S \in S^k$, the following holds:

$$v_{mod}(S \cup \{i\}) - v_{mod}(S) = \alpha \frac{l_{in}(S \cup \{i\})}{m} - \beta \frac{l_{out}(S \cup \{i\})}{m} - \alpha \frac{l_{in}(S)}{m} + \beta \frac{l_{out}(S)}{m},$$

which is equal to

$$\frac{\alpha}{m}[l_{in}(S \cup \{i\}) - l_{in}(S)] - \frac{\beta}{m}[l_{out}(S \cup \{i\}) - l_{out}(S)] = \frac{\alpha}{m}k - \frac{\beta}{m}(l_i - 2k).$$

From previous equation, we can rewrite the semivalue as:

$$\Psi_i(v_{mod}) = \sum_{k=0}^{l_i} \sum_{S \in S^k} P(S)\left(\frac{\alpha}{m}k - \frac{\beta}{m}(l_i - 2k)\right).$$

Now, as $\displaystyle\sum_{S \in S^k} P(S) = \sum_{S \in S^{(l_i-k)}} P(S)$ (the constraint that we impose to the semivalue) we have:

- If $l_i$ is odd,

$$\Psi_i(v_{mod}) = \sum_{k=0}^{(l_i-1)/2} \sum_{S \in S^k} P(S)\left(\frac{\alpha}{m}k - \frac{\beta}{m}(l_i - 2k) + \frac{\alpha}{m}(l_i - k) - \frac{\beta}{m}(l_i - 2(l_i - k))\right),$$

which is equal to

$$\sum_{k=0}^{(l_i-1)/2} \sum_{S \in S^k} P(S)\frac{\alpha l_i}{m} = \frac{\alpha l_i}{m} \sum_{k=0}^{(l_i-1)/2} \sum_{S \in S^k} P(S).$$

Now, taking into account that $\sum_{k=0}^{l_i}\sum_{S\in S^k}P(S)=1$ and $\sum_{S\in S^k}P(S)=\sum_{S\in S^{(l_i-k)}}P(S)$, then the following holds

$$\sum_{k=0}^{(l_i-1)/2}\sum_{S\in S^k}P(S)=\frac{1}{2}$$

and thus

$$\Psi_i(v_{mod})=\alpha\frac{l_i}{2m}.$$

- If $l_i$ is even, then $\Psi_i(v_{mod})$ can be rewritten as:

$$\sum_{k=0}^{(l_i-2)/2}\sum_{S\in S^k}P(S)\left(\frac{\alpha}{m}k-\frac{\beta}{m}(l_i-2k)+\frac{\alpha}{m}(l_i-k)-\frac{\beta}{m}(l_i-2(l_i-k))\right)+$$

$$+\sum_{S\in S^{l_i/2}}P(S)\left(\frac{\alpha}{m}(l_i-\frac{l_i}{2})-\beta\frac{l_i-2(l_i/2)}{m}\right),$$

which is equal to

$$\sum_{k=0}^{(l_i-1)/2}\sum_{S\in S^k}P(S)\frac{\alpha l_i}{m}+\sum_{S\in S^{l_i/2}}P(S)\frac{\alpha l_i}{2m}=\frac{\alpha l_i}{m}\left(\sum_{k=0}^{(l_i-1)/2}\sum_{S\in S^k}P(S)+\frac{1}{2}\sum_{S\in S^{(l_i/2)}}P(S)\right).$$

Now, taking into account that

$$\sum_{k=0}^{l_i}\sum_{S\in S^k}P(S)=1\quad\text{and}\quad\sum_{S\in S^k}P(S)=\sum_{S\in S^{(l_i-k)}}P(S)$$

then the following holds

$$\sum_{k=0}^{(l_i-2)/2}\sum_{S\in S^k}P(S)+\frac{1}{2}\sum_{S\in S^{(l_i/2)}}P(S)=\frac{1}{2}$$

and thus

$$\Psi_i(v_{mod})=\frac{\alpha l_i}{2m}.$$

$\square$

In the following two propositions we see that the Shapley and the Banzahf-Coleman values are particular cases of previous situation.

**Proposition 2.** *Let $(N,v_{mod})$ denote the linear modularity game and let $i$ be a node in N, then the Shapley value of this node i is:*

$$Sh_i(v_{mod})=\alpha\frac{l_i}{2m}.$$

**Proof.** Because of the results demonstrated in the Proposition 1, we only have to demonstrate that

$$\sum_{S\in S^k}\frac{|S|!(|N|-1-|S|)!}{|N|!}=\sum_{S\in S^{(l_i-k)}}\frac{|S|!(|N|-1-|S|)!}{|N|!}.$$

For each $S\in S^k$, $N\setminus(S\cup\{i\})$ belong to $S^{l_i-k}$ and viceversa. Then, the the cardinal of $S^k$ is the same that the cardinal of $S^{l_i-k}$. Therefore, a sufficient condition to prove previous equality is that $P(S)=P(N\setminus(S\cup\{i\}))$ where $P(S)=\frac{|S|!(|N|-1-|S|)!}{|N|!}$.

Effectively, $P(S)=\frac{|S|!(|N|-1-|S|)!}{|N|!}=\frac{(|N|-1-|S|)!(|N|-1-(|N|-1-|S|))!}{|N|!}=P((N\setminus(S\cup\{i\})))$. $\square$

**Proposition 3.** *Let* $(N, v_{mod})$ *denote the linear modularity game and let i denote a node in N. Then Banzahf-Coleman value of this node i is:*

$$B_i(v_{mod}) = \alpha \frac{l_i}{2m}$$

**Proof.** From Proposition 1, we only have to check the following equation.

$$\sum_{S \in S^k} \frac{1}{2^{|N|-1}} = \sum_{S \in S^{(l_i-k)}} \frac{1}{2^{|N|-1}}. \tag{9}$$

Equation (9) trivially holds since the cardinal of $S^k$ is equal to the cardinal of $S^{l_i-k}$.

On the following, we will refer to the node-weighted SP betweenness measure as the node-game SP betweenness measure. Also, we will use the terms power, Shapley value or Banzahf-Coleman in a similar way since these concepts coincide for the linear modularity game. □

**Example 6.** *Consider the graph defined in Example 3. From a centrality point of view, nodes 1 and 6 have low centrality. Nodes 2 and 5 have high centrality and nodes 3 and 4 have medium centrality. Considering the power solution given by the linear modularity game in the graph and using the minimum operator, we obtain the following weights matrix:*

$$H = (h_{rs}) = \frac{\alpha}{12} \begin{pmatrix} 1 & 1 & 1 & 1 & 1 & 1 \\ 1 & 3 & 2 & 2 & 3 & 1 \\ 1 & 2 & 2 & 2 & 2 & 1 \\ 1 & 2 & 2 & 2 & 2 & 1 \\ 1 & 3 & 2 & 2 & 3 & 1 \\ 1 & 1 & 1 & 1 & 1 & 1 \end{pmatrix}.$$

*Intermediate links in the SP communication between nodes 2 and 5 have greater betweenness, so the divisive algorithm will try to separate them. It is not difficult to see that the node-game SP betweenness for matrix H will rank the links of the graph in the following way: $w_{2,3} = w_{2,4} = w_{3,5} = w_{4,5} > w_{1,2} = w_{5,6}$. This new rank fixes the poor performance of the classical SP betweenness. Without lost of generality, let us suppose that the first link we cut is the link $(2, 4)$ or $(3, 5)$ (the other situation is analogous). Then, the first cut of the algorithm will partition the graph into two clusters: $C_1 = \{1, 2, 3\}$ and $C_2 = \{4, 5, 6\}$, with modularity $Q = \frac{1}{6}$.*

**Remark 1.** *Let us observe that the value $\alpha$ does not affect the ranking of the edges in terms of node-game SP betweeness measure if $\alpha > 0$. So, without loss of generality, we will consider from now and on $\alpha = 1$.*

## 4. Computational Complexity

The previous section revealed how the inclusion of the importance or power of the nodes when calculating SP betweenness improves the performance of the *GN* algorithm. We will see that this new algorithm does not increases the computational complexity of the *GN* algorithm.

As it is pointed in [1], the betweenness of all edges of the graph can be calculated in a time that scales as $O(mn)$ (or $O(n^2)$ on a sparse graph). Taking into account that in the *GN* algorithm this calculation is repeated $m$ times, the computational complexity of the *GN* algorithm is $O(m^2n)$ (or $O(n^3)$ on a sparse networks).

The main difference between the calculation of betweenness and node-game betweenness for all edges of the network is the calculation of the power for all nodes. So, if we denote by $f(n)$ the computational complexity associated with its calculation, the complexity associated with the calculation of the node-game betweenness for all edges is $O(mn + f(n))$. As this process is repeated $m$ times, the complexity of new algorithm (denoted as GICE algorithm) for a general power measure is $O(m^2n + mf(n))$. Let us note that the Shapley

value of the linear modularity game can be obtained in order $O(m)$, and it is not necessary to recalculate in each iteration as the power of a node can be update automatically after one edge is removed from the network. Hence, the computational complexity of our algorithm is $O(m^2n + m)$. As $m$ is bounded by $m^2n$, we can say that this algorithm has a complexity of $O(m^2n)$ (or $O(n^3)$ on sparse networks), when the power measure is the Shapley value of the linear modularity game.

## 5. Reducing the Computational Complexity of the Node-Game Betweeness Measure

The computational complexity of the *GN* algorithim, whose order is $O(m^2n)$ or $O(n^3)$ in sparse networks, is one of the main inconveniences of this method. Hence in practice, it is not possible to analyze large networks related to real problems. To be competitive in large networks, several authors (see [9,39,40] among others) have developed faster techniques for calculating the SP betweenness measure for all the edges of a graph. For example, in [40], it is proposed a modification of the *GN* algorithm to improves the speed of SP edge betwenness measure calculation. The computational complexity of the calculation of the SP betweeness (with order $O(nm)$) is drastically reduced: only some key nodes, which are defined a priori and called gene co-ocurrences, are used. This SP betweeness approximation is obtained by considering the shortest paths from some key nodes to all nodes. Using a breadth first search, the calculation of the SP edge betweeness when considering the SP from one center to the rest of the nodes, has a linear complexity ($O(m)$) in unweighted graphs. The calculation of the SP edge betweenness for every adjacent nodes may be reduced from $O(nm)$ to ($O(f(n)m)$ when there are $f(n) < n$ nodes in the networks. For instance, when $f(n) = \log(n)$, the computational complexity related to the calculation of the SP edge betweenness approximation is roughly $O(m)$. In this scenario, the complexity of the *GN* algorithm can be reduced to $O(m^2)$ or $O(n^2)$ in sparse networks.

Obviously, the selection of the $f(n)$ nodes in the network is essential in the performance of the divisive algorithm. In [39], the key nodes are chosen at random, deriving a sort of Monte Carlo estimate. The authors indicate that, for each connected subgraph, it is sufficient to choose a number of centers growing as the logarithm of the number of vertices of the component. Obviously, the partitions are, in general, different for different choices of the set of center vertices.

Following the same spirit of [39,40], in this paper we propose a fast version of the algorithm presented in Section 3, by reducing the calculation of the node-game SP betweeness measure in a similar way. When only considering the $\log(n)$ nodes with maximum power in the whole graph to approach the node-game SP betweenness measure, the corresponding computational complexity of this approximation has an order of $O(m\log(n))$. Taking into account that this calculation has to be done $m$ times, we can conclude that the computational complexity associated with this *fast* version is $O(m^2\log(n))$, that for sparse networks is $O(n^2)$ (the same complexity as the CNM algorithm). It allows us to analyze large networks. In Table 1, we present the computational complexity associated with the different algorithms that have been used in this paper. These are a representative sample of the algorithm that can obtain a hierarchical clustering of the network. Let us note that for hierarchical clustering algorithms as they are defined in Section 3, the length of the dendrogram (usually denote as $d$) is $n$, as we need to obtain all the partitions of the graph and, in each iteration, we reduce (in agglomerative methods) or increase (in divisive) the number of groups in one unit.

**Table 1.** Computational complexity of the GN, CNM, Walktrap, N2012, Radicchi, GICE and its fast version (GICEF) algorithms.

| Algorithm | Computational Complexity | Order on Sparse Networks |
|:---:|:---:|:---:|
| GN | $O(m^2 n)$ | $O(n^3)$ |
| CNM | $O(mdlog(n)) = O(mnlog(n))$ | $O(n^2 log(n)) \sim O(n^2)$ |
| Walktrap | $O(mn^2)$ | $O(n^3)$ |
| N2012 | $O(n^2 log(n))$ | $O(n^2 log(n)) \sim O(n^2)$ |
| Radicchi | $O(m^4 / n^2)$ | $O(n^2)$ |
| GICE | $O(m^2 n)$ | $O(n^3)$ |
| GICEF | $O(m^2 log(n))$ | $O(n^2 log(n)) \sim O(n^2)$ |

## 6. Computational Results in Real Well-Known Examples

When you want to test the effectiveness of an algorithm, it is necessary to answer several questions. The first one is related to those algorithms with which you have to compare yours. The second is about the selection of the examples in which you test your effectiveness. The third is related to the accuracy measures that show the quality of the output of the different algorithms. Then we give an answer for these questions to validate the performance of our algorithm.

In Section 2.2 we give a short review about some of the most well-known techniques that can deal with the hierarchical clustering network problem: *GN*, *CNM*, *Walktrap*, *N*2012 and *Radicchi*. We compare our algorithm with these methods.

Regarding the selection of the examples, in the field of community detection problems there exist two types of benchmark examples ([22,41]). The first type of data sets are some well-known examples as *the karate club network, the dolphins network, Les miserables network, the football network, the centrality authors network* or *the jazz network* among others, in which most of the clustering networks have been tested. The second class of test examples are generated by simulation with an *a priori* knowledge of the classes. The simulation model starts with a known number of classes, and it assigns different probabilities for two nodes of being connected in the same community ($p_{in}$) or in different communities ($p_{out}$). Generating the networks in this way, we can test the performance of an algorithm that searches an optimal partition measuring how close or similar are the partition given by an algorithm with the a priori classes of the network.

Finally, we focus on the accuracy measures used to compare the results given by two different algorithms. Apart from some innate features of any algorithm, as its complications to implement or the computational complexity associated with it, there are other two problems which have to be faced:

1. The comparison of two partitions of a graph.
2. The comparison of two sequence of partitions.

Although there exist others accuracy measures ([1]), in this paper we use the concept of modularity to measure the quality of a partition in a graph. The modularity represents the homogeneity in terms of cluster density, and punishes the relations that exist among groups. Nevertheless, in some situations, the modularity of two partitions are very similar, so it may be necessary to consider more information to discriminate them. Thus, another criterion considered in the clustering literature is the size homogeneity. In our opinion, it is clear that when two partitions are very similar it would be better to have homogeneous groups in terms of size.

Given a partition $P = \{C_1, \ldots C_r\}$ of a set of nodes $N$ with size vector $S = (s_1, \ldots s_r)$, many measures of heterogeneity in groups are available. Since we use this measure to compare the dispersion of two samples with different average and the sum of the size vector is always constant, we use the coefficient of variation, which can be formalized as :

$$CV_S = \frac{\sigma_S}{\bar{s}} \qquad (10)$$

where $\bar{s}$ is the average size and $\sigma_S$ is the typical deviation for sample $s_1, \ldots s_r$.

As we have done with the modularity vector, given a hierarchical clustering of a graph $\mathcal{D} = (P_1, P_2, \ldots, P_{n-k})$, we denote by $\mathcal{CV}(\mathcal{D}) = (CV_1(D), CV_2(D), \ldots, CV_{n-k}(D))$ the coefficient of variation vector associated to the dendrogram $D$.

Despite the existence of measures to determine the quality of a partition (in this case we use the modularity measure), it is not clear how to measure the goodness of a hierarchical clustering. The problem of comparing dendrograms is an open topic which requires further study. A possible criterion for comparing two dendrograms [2,6] is to compare the partitions obtained by hierarchical clustering in the step in which maximum modularity is reached. However, we describe an example to show how this criterion could be unfair when two dendrograms are compared.

**Example 7.** *In this example we consider a chain $C = (N, E)$ with 12 nodes, connected as it is represented in the Figure 4. In this scenario, the maximum of modularity is achieved with the partition $P = \{C_1 = \{1, 2, 3, 4\}, C_2 = \{5, 6, 7, 8\}$ and $C_3 = \{9, 10, 11, 12\}\}$.*

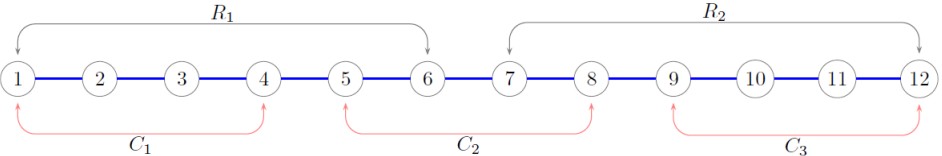

**Figure 4.** Chain $Ch = (N, E)$ of 12 nodes.

*However, if we analyze the algorithm from a divisive point of view, the first obtained partition should be $P^R = \{R_1 = \{1, 2, 3, 4, 5, 6\}, R_2 = \{7, 8, 9, 10, 11, 12\}\}$, in which it is reached the maximum of modularity in two clusters. Nevertheless, pay attention to the fact that, if $P^R$ were considered the first partition, the optimal partition $P = \{C_1, C_2, C_3\}$ will never be reached by any divisive method. Even so, that optimal partition $P$ with three clusters $C_1, C_2, C_3$ may be reached with a first iteration such as $\{1, 2, 3, 4\}$ and $\{5, 6, 7, 8, 9, 10, 11, 12\}$.*

Note that divisive hierarchical clustering methods presents the evolution of a group which is divided into a network. Then, we should not take into consideration just a specific stage, even though the initial iterations were considered to be more relevant than the followings.

In our opinion, the comparison of two hierarchical clusters has to take into account more than just one step because a these methods not only provide the obtained final partition, but they provide the whole evolution of a group split into a network. Hence, we propose some different criteria to compare dendrograms, assuming that the problem of comparing partitions has been solved by the modularity definition (and in case of similar modularity value by its homogeneity).

Given a dendrogram $\mathcal{D}$, we denote by $t_{max}(D)$ the iteration in which the maximum modularity is reached, i.e., $t_{max}(D) = arg\{ Max_{1 \leq t \leq n-k} Q_t(D)\}$.

- Maximum modularity criterion (denoted by $Cr_1$). To compare two (or more) dendrograms, the partitions obtained for each dendrogram are compared in the step in which they reach maximum modularity. Although this criterion presents the problems described above, $Cr_1$ is a unique criterion that can be derived from the literature on social network clustering to compare two divisive hierarchical clustering algorithms. It could be used to determine a possible number of clusters/groups in a network for any further optimization partitional methods that require this information. $Cr_1$ can be defined as a function from the set of all possible dendrograms to the set of real numbers $Cr_1 : \mathcal{P}(\mathcal{D}) \rightarrow \mathbb{R}$ such that:

$$Cr_1(D) = Q_{t_{max}(D)}(D)$$

- Average criterion. To compare two (or more) dendrograms, this compares the average of the modularity vector. In our opinion, this criterion has too many inconveniences. The most important one is that this criterion gives the same importance to the first partition as to the final one. It is clear that when a network is connected, the first cuts have more relevance than the last ones, in which only a few nodes are connected. These last iterations are usually not analyzed. Formally this criteria can be defined as a function from the set of all possible dendrograms to the set of real numbers $Cr : \mathcal{P}(\mathcal{D}) \rightarrow \mathbb{R}$ such that:

$$Cr(D) = \sum_{t=1}^{n-k} \frac{Q_t(D)}{n-k}$$

- Lexicographical criterion (denoted by $Cr_2$). One way to compare two (or more) dendrograms, is comparing the modularity vectors lexicographically (i.e., the first partitions are compared; if these are equal, then the second partitions are compared, and so on). This criterion leads to greater importance for the first partition than for subsequent ones. Formally $Cr_2$ establish an order in the set of $\mathcal{P}(\mathcal{D})$ in such a way $D \widetilde{<} D'$ iff the modularity vector $\mathcal{Q}_\mathcal{D}$ is lexicographically lower than $\mathcal{Q}_{\mathcal{D}'}$ (i.e $D \widetilde{<} D'$ iff there exists $t$ such that $Q_r(D) = Q_r(D')$ for $r < t$ and $Q_t(D) < Q_t(D')$ ).
- Average truncated by the maximum criterion (denoted by $Cr_3$). Following the average criterion but fixing the problem of considering all the partitions in the dendrogram, this criterion only compares the dendrograms until they reach a maximum value. This approach follows an idea in which the hierarchical process stops at the step in which the maximum (in terms of modularity) is reached. Formally, this criteria can be defined as a function from the set of all possible dendrograms to the set of real numbers $Cr_3 : \mathcal{P}(\mathcal{D}) \rightarrow \mathbb{R}$ such that:

$$Cr_3(D) = \sum_{t=1}^{t_{max}(D)} \frac{Q_t(D)}{t_{max}(D)}$$

Similarly, the size homogeneity of a given partition could be used to establish different criteria to discriminate between two dendrograms. We suggest the following homogeneity criteria.

- Maximum modularity homogeneity (denoted by $SCr_1$). Following the ideas described for $Cr_1$, we compare two or more dendrograms by comparing the homogeneity of the partitions obtained in the step in which the maximum modularity is reached. Formally, this criteria can be defined as a function from the set of all possible dendrograms to the set of real numbers $SCr_1 : \mathcal{P}(\mathcal{D}) \rightarrow \mathbb{R}$ such that:

$$SCr_1(D) = CV_{t_{max}(D)}(D)$$

- Lexicographical criterion (denoted by $SCr_2$). We compare the homogeneity vectors lexicographically. This criterion assigns greater importance to the first partitions than to subsequent ones. Formally $SCr_2$ establishes an order of the set of $\mathcal{P}(\mathcal{D})$ in such a way $D \widetilde{<} D'$ iff the $\mathcal{CV}_\mathcal{D}$ is lexicographically lower than $\mathcal{CV}_{\mathcal{D}'}$ (i.e $D \widetilde{<} D'$ iff there exist $t$ such that $CV_r(D) = CV_r(D')$ for $r < t$ and $CV_t(D) < CV_t(D')$ ).
- Homogeneity criterion for the average truncated by the maximum (denoted by $SCr_3$). This criterion only compares the dendrograms until the step in which the maximum modularity is reached. Formally this criteria can be defined as a function from the set of all possible dendrograms to the set of real numbers $SCr_3 : \mathcal{P}(\mathcal{D}) \rightarrow \mathbb{R}$ such that:

$$SCr_3(D) = \sum_{t=1}^{t_{max}(D)} \frac{CV_t(D)}{t_{max}(D)}$$

Taking into account previous considerations, we analyze the performance of our algorithm in four well-known examples, comparing with the five hierarchical algorithms *GN*, *CNM*, *Walktrap*, *N2012 and Radicchi*, in the criteria $Cr_1, Cr_2, Cr_3$ and $SCr_1, SCr_2, SCr_3$:

1.  The *karate club* network [42]. The nodes of this graph represent the partners of a karate club, and every edge connecting two nodes represents they have a friendship relation. Historically, cutting the network into two groups has been studied because of its sociological importance. This network has 34 nodes and 156 links. We show the performance of the different algorithms in Table 2.
2.  The *dolphins* network [43]. This network is an undirected social network of frequent associations between some dolphins in a community living off Doubtful Sound, New Zealand. It comprises 62 nodes and 159 links. We show the performance of the different algorithms in Table 3.
3.  *Les Miserables* network [44]. The nodes of this network are the actors that appear in the book by *Victor Hugo*, and the links represent the relations among them. The network has 77 nodes and 254 links. We show the performance of the different algorithms in Table 4.
4.  The *centrality authors* [45], taken from the pajek web page. This network is related to several papers about network centrality, and how these work are interconnected between them by cross-reference, considering the cites between 1940 and 1979. Every node represents a paper, so the edges represent if there is a citation among the connected nodes 129 nodes and 613 links. Let us note that the result of Walktrap algorithm for this network is not included, as this method can not deal with non-connected networks. We show the performance of the different algorithms in Table 5.

Finally, to summarize both criteria (Modularity and Homogeneity) into one (from the perspectives $Cr_1, Cr_2$ and $Cr_3$), we have considered that modularity is more important than homogeneity. It is clear that if you have a completed graph with 25 nodes connected with another completed graph with 100 nodes, the optimal partition should have two clusters with sizes 25 and 100. Obviously, there are others partitions with more size homogeneity, but their modularity will be significantly lower. Nevertheless, homogeneity criteria should have an impact in the quality of a partition when the modularity values are very similar or are not significantly different. As we have pointed out previously, from our point of view, it is clear that two partitions $P$ and $P'$ with similar modularity values (for example $Q(P) = 0.398$ and $Q(P') = 0.388$) but with sizes $s(P) = (80, 20)$ and $s(P') = (51, 49)$ are entirely different.

**Table 2.** Modularity and Homogeneity for partitions with $k$ groups ($k = 2, \ldots, 6$) obtained using the GN, CNM, Walktrap, N2012, Radicchi, our GICE and its fast version (GICEF) for the karate club network. The criteria values are shown in parentheses.

| Modularity Groups | GN | CNM | Walktrap | N2012 | Radicchi | GICE | GICEF |
|---|---|---|---|---|---|---|---|
| 2 | 0.360 | 0.358 | 0.335 | **0.371** | 0.132 | 0.360 | 0.360 |
| 3 | 0.348 | 0.371 | 0.343 | 0.141 | 0.112 | 0.391 | 0.374 |
| 4 | 0.363 | **0.388** | 0.352 | 0.169 | **0.354** | 0.406 | **0.391** |
| 5 | **0.401** | 0.380 | **0.353** | 0.183 | 0.353 | **0.406** | 0.384 |
| 6 | 0.392 | 0.362 | 0.344 | 0.171 | 0.316 | 0.362 | 0.347 |
| **Cr1** | 0.40(2) | 0.39(4) | 0.35(7) | 0.37(5) | 0.35(6) | 0.41(1) | 0.39(3) |
| **Cr2** | 4 | 5 | 6 | 1 | 7 | 2 | 3 |
| **Cr3** | 0.37(5) | 0.37(3) | 0.35(6) | 0.37(4) | 0.20(7) | 0.39(1) | 0.38(2) |
| **Homogeneity Groups** | GN | CNM | Walktrap | N2012 | Radicchi | GICE | GICEF |
| 2 | 0.118 | 0.050 | 0.176 | **0.058** | 0.705 | 0.118 | 0.118 |
| 3 | 0.654 | 0.324 | 0.560 | 0.463 | 1.103 | 0.244 | 0.527 |
| 4 | 0.746 | **0.306** | 0.555 | 0.489 | **0.800** | 0.263 | **0.476** |
| 5 | **0.569** | 0.352 | **0.300** | 0.717 | 0.796 | **0.455** | 0.459 |
| 6 | 0.703 | 0.474 | 0.474 | 0.897 | 0.892 | 0.437 | 0.476 |
| **SCr1** | 0.57(6) | 0.31(3) | 0.30(2) | 0.06(1) | 0.80(7) | 0.46(4) | 0.48(5) |
| **SCr2** | 5 | 1 | 6 | 2 | 7 | 3 | 4 |
| **SCr3** | 0.52(6) | 0.23(2) | 0.40(5) | 0.06(1) | 0.87(7) | 0.27(3) | 0.37(4) |

**Table 3.** Modularity and Homogeneity for partitions with *k* groups (*k* = 2, . . . , 7) obtained using the GN, CNM, Walktrap, N2012, Radicchi, our GICE and its fast version (GICEF) for the dolphins network. The criteria values are shown in parentheses.

| Modularity Groups | GN | CNM | Walktrap | N2012 | Radicchi | GICE | GICEF |
|---|---|---|---|---|---|---|---|
| 2 | 0.378 | 0.392 | 0.390 | 0.347 | 0.373 | 0.384 | 0.380 |
| 3 | 0.381 | 0.436 | 0.385 | 0.344 | 0.430 | 0.473 | **0.478** |
| 4 | 0.458 | **0.507** | **0.489** | 0.343 | 0.430 | 0.455 | 0.448 |
| 5 | **0.519** | 0.502 | 0.488 | 0.339 | **0.502** | 0.467 | 0.474 |
| 6 | 0.514 | 0.500 | 0.485 | 0.434 | 0.485 | **0.490** | 0.466 |
| 7 | 0.518 | 0.492 | 0.472 | **0.460** | 0.464 | 0.469 | 0.460 |
| Cr1 | 0.52(1) | 0.51(2) | 0.49(5) | 0.46(7) | 0.50(3) | 0.49(4) | 0.48(6) |
| Cr2 | 5 | 1 | 2 | 7 | 6 | 3 | 4 |
| Cr3 | 0.43(3) | 0.45(2) | 0.42(6) | 0.38(7) | 0.43(4) | 0.45(1) | 0.43(5) |
| Homogeneity Groups | GN | CNM | Walktrap | Ň | Radicchi | GICE | GICEF |
| 2 | 0.323 | 0.096 | 0.258 | 0.387 | 0.355 | 0.290 | 0.322 |
| 3 | 0.731 | 0.661 | 0.730 | 0.812 | 0.593 | 0.280 | **0.348** |
| 4 | 0.761 | **0.548** | **0.529** | 1.022 | 0.772 | 0.411 | 0.523 |
| 5 | **0.592** | 0.435 | 0.560 | 1.229 | **0.582** | 0.553 | 0.524 |
| 6 | 0.708 | 0.322 | 0.667 | 1.068 | 0.611 | **0.409** | 0.714 |
| 7 | 0.752 | 0.435 | 0.760 | **0.804** | 0.757 | 0.364 | 0.926 |
| SCr1 | 0.59(6) | 0.55(4) | 0.53(3) | 0.80(7) | 0.58(5) | 0.41(2) | 0.35(1) |
| SCr2 | 5 | 1 | 2 | 7 | 6 | 3 | 4 |
| SCr3 | 0.60(6) | 0.44(3) | 0.51(4) | 0.89(7) | 0.58(5) | 0.39(2) | 0.34(1) |

**Table 4.** Modularity and Homogeneity for partitions with *k* groups (*k* = 2, . . . , 11) obtained using the GN, CNM, Walktrap, N2012, our GICE and its fast version (GICEF) for *Les miserables* network. The criteria values are shown in parentheses.

| Modularity Groups | GN | CNM | Walktrap | N2012 | Radicchi | GICE | GICEF |
|---|---|---|---|---|---|---|---|
| 2 | 0.075 | 0.369 | 0.075 | 0.000 | 0.373 | 0.376 | 0.361 |
| 3 | 0.260 | 0.442 | 0.081 | 0.006 | 0.480 | 0.431 | 0.391 |
| 4 | 0.266 | **0.457** | 0.217 | 0.191 | 0.502 | 0.528 | 0.487 |
| 5 | 0.415 | 0.446 | 0.474 | 0.444 | 0.498 | 0.538 | 0.530 |
| 6 | 0.459 | 0.447 | 0.476 | **0.465** | 0.469 | **0.550** | **0.530** |
| 7 | 0.455 | 0.455 | 0.508 | 0.449 | 0.471 | 0.535 | 0.528 |
| 8 | 0.453 | 0.452 | **0.521** | 0.447 | 0.471 | 0.532 | 0.523 |
| 9 | 0.452 | 0.450 | 0.519 | 0.439 | **0.509** | 0.524 | 0.520 |
| 10 | 0.452 | 0.448 | 0.511 | 0.444 | 0.505 | 0.516 | 0.513 |
| 11 | **0.538** | 0.446 | 0.517 | 0.440 | 0.498 | 0.509 | 0.495 |
| Cr1 | 0.54(2) | 0.46(7) | 0.52(4) | 0.47(6) | 0.51(5) | 0.55(1) | 0.53(3) |
| Cr2 | 5 | 3 | 6 | 7 | 2 | 1 | 4 |
| Cr3 | 0.38(5) | 0.42(4) | 0.34(6) | 0.22(7) | 0.47(2) | 0.48(1) | 0.46(3) |
| Homogeneity Groups | GN | CNM | Walktrap | N2012 | Radicchi | GICE | GICEF |
| 2 | 0.740 | 0.403 | 0.740 | 0.974 | 0.039 | 0.060 | 0.511 |
| 3 | 0.863 | 0.640 | 1.091 | 1.146 | 0.397 | 0.415 | 0.712 |
| 4 | 1.086 | **0.773** | 1.171 | 1.175 | 0.373 | 0.354 | 0.471 |
| 5 | 0.976 | 0.918 | 0.910 | 0.894 | 0.532 | 0.265 | 0.407 |
| 6 | 0.627 | 0.988 | 0.980 | **0.790** | 0.687 | **0.350** | **0.445** |
| 7 | 0.726 | 1.066 | 0.906 | 0.530 | 0.807 | 0.395 | 0.569 |
| 8 | 0.820 | 1.157 | **0.736** | 0.649 | 0.898 | 0.307 | 0.736 |
| 9 | 0.896 | 1.232 | 0.764 | 0.750 | **0.678** | 0.362 | 0.836 |
| 10 | 0.917 | 1.295 | 0.818 | 0.819 | 0.779 | 0.394 | 0.951 |
| 11 | **0.703** | 1.340 | 0.466 | 0.833 | 0.791 | 0.460 | 0.946 |
| SCr1 | 0.70(4) | 0.77(6) | 0.74(5) | 0.79(7) | 0.68(3) | 0.35(1) | 0.45(2) |
| SCr2 | 5 | 3 | 6 | 7 | 1 | 2 | 4 |
| SCr3 | 0.82(5) | 0.61(4) | 0.93(6) | 1.00(7) | 0.55(3) | 0.29(1) | 0.51(2) |

**Table 5.** Modularity and Homogeneity for partitions with $k$ groups ($k = 13, \ldots, 60$) obtained using the GN, CNM, N2012, Radicchi, our GICE and its fast version (GICEF) for the authors network. The criteria values are shown in parentheses.

| Modularity Groups | GN | CNM | N2012 | Radicchi | GICE | GICEF |
|---|---|---|---|---|---|---|
| 13 | 0.012 | 0.073 | 0.000 | 0.025 | 0.033 | 0.104 |
| 14 | 0.025 | 0.143 | 0.000 | 0.131 | 0.078 | 0.157 |
| 15 | 0.025 | 0.221 | 0.002 | 0.156 | 0.161 | 0.229 |
| 16 | 0.026 | 0.246 | 0.002 | 0.171 | 0.224 | 0.232 |
| 17 | 0.027 | **0.246** | 0.002 | 0.240 | 0.240 | 0.245 |
| 18 | 0.029 | 0.245 | 0.116 | 0.258 | 0.238 | **0.250** |
| 19 | 0.031 | 0.245 | 0.115 | 0.255 | 0.247 | 0.231 |
| 20 | 0.032 | 0.244 | 0.113 | 0.255 | **0.262** | 0.225 |
| 21 | 0.034 | 0.243 | 0.111 | 0.258 | 0.260 | 0.226 |
| 22 | 0.035 | 0.242 | **0.177** | **0.260** | 0.258 | 0.221 |
| . . . | . . . | . . . | . . . | . . . | . . . | . . . |
| 60 | **0.180** | . . . | . . . | . . . | . . . | . . . |
| **Cr1** | 0.18(5) | 0.25(4) | 0.18(6) | 0.26(2) | 0.26(1) | 0.25(3) |
| **Cr2** | 5 | 2 | 6 | 4 | 3 | 1 |
| **Cr3** | 0.09(5) | 0.19(3) | 0.06(6) | 0.20(2) | 0.19(4) | 0.20(1) |

| Homogeneity Groups | GN | CNM | N2012 | Radicchi | GICE | GICEF |
|---|---|---|---|---|---|---|
| 13 | 3.029 | 2.863 | 3.000 | 3.000 | 2.860 | 2.989 |
| 14 | 3.065 | 1.743 | 3.096 | 2.650 | 2.560 | 2.568 |
| 15 | 3.153 | 1.552 | 3.153 | 2.487 | 2.209 | 1.895 |
| 16 | 3.234 | 1.402 | 3.108 | 2.394 | 1.940 | 1.662 |
| 17 | 3.310 | **1.427** | 2.989 | 1.767 | 1.800 | 1.428 |
| 18 | 3.380 | 1.477 | 2.359 | 1.584 | 1.707 | **1.238** |
| 19 | 3.446 | 1.514 | 2.395 | 1.591 | 1.493 | 1.387 |
| 20 | 3.507 | 1.549 | 2.463 | 1.423 | **1.196** | 1.224 |
| 21 | 3.564 | 1.580 | 2.511 | 1.415 | 1.025 | 1.057 |
| 22 | 3.814 | 1.617 | **2.207** | **1.404** | 1.017 | 1.246 |
| . . . | . . . | . . . | . . . | . . . | . . . | . . . |
| 60 | **3.006** | . . . | . . . | . . . | . . . | . . . |
| **SCr1** | 3.01(6) | 1.43(4) | 2.21(5) | 1.40(3) | 1.20(1) | 1.24(2) |
| **SCr2** | 6 | 2 | 5 | 4 | 1 | 3 |
| **SCr3** | 3.12(6) | 1.80(1) | 2.73(5) | 2.00(4) | 1.97(3) | 1.96(2) |

To measure the differences between two partitions we could used the absolute value $|Q(P) - Q(P')| = |0.398 - 0.388| = 0,01$ (i.e the partition $P$ improved the modularity of $P'$ in $0,01$), or, the relative value, $\frac{|Q(P) - Q(P')|}{Q(P')} = \frac{|0.398 - 0.388|}{0.388} = 0,025$ (i.e., the modularity of $P'$ is improved in a 2.5% by the partition $P$). Taking into account that the homogeneity and modularity take values in different scales, we have used the second expression (the relative value) to compare two dendrograms.

Following $(Cr_1, SCr_1)$, in Tables 6–9 we have summarized the modularity and homogeneity criteria into one as we have described previously. Formally, given two hierarchical partitions $D$ and $D'$ and an error $\epsilon$ value, we have considered that $D$ is greater than $D'$ (or is better) in the criteria $(Cr_1, SCr_1)$ if and only if one of the two cases holds:

- $\frac{Cr_1(D) - Cr_1(D')}{Cr_1(D')} > \epsilon$ (i.e., the percentage of gain in the modularity criteria of maximum in $D$ is significatively better than $D'$) or
- $\frac{|Cr_1(D) - Cr_1(D')|}{min\{Cr_1(D), Cr_1(D')\}} \leq \epsilon$ and $\frac{SCr_1(D') - SCr_1(D)}{SCr_1(D)} > \epsilon$ (i.e., if both hierarchical partitions are similar in terms of maximum modularity then the homogeneity criteria has to be significatively better).

Following $(Cr_2, SCr_2)$, in Tables 6–9 we have combined the modularity and homogeneity criteria. Formally, given two hierarchical partitions $D$ and $D'$, we have considered that $D$ is greater than $D'$ (or is better) in the criteria $(Cr_2, SCr_2)$ if and only if one of the following cases holds:

- There exists $l \in \{1, \ldots, n\}$ such that $\frac{|Q(P_r) - Q(P'_r)|}{min\{Q(P_l), Q(P'_l)\}} \leq \epsilon$ for all $r < l$ and $\frac{Q(P_l) - Q(P'_l)}{Q(P'_l)} > \epsilon$ (i.e., D and D' are significantly equals in the first $l - 1$ partitions but the modularity of $D$ is significantly better in the $l$ partition) or

- $\frac{|Q(P_i) - Q(P'_i)|}{min\{Q(P_i), Q(P'_i)\}} \leq \epsilon$ for all $i$ and the homogeneity is lexicographically better in a similar way as we have done in the previous step for the modularity.

Following $(Cr_3, SCr_3)$, in Tables 6–9 we have combined the modularity and homogeneity criteria into one. Formally, given two hierarchical partitions $D$ and $D'$ and an error $\epsilon$ value, we have considered that $D$ is greater than $D'$ (or is better) in the criteria $(Cr_3, SCr_3)$ if and only if one of the two cases holds.

- $\frac{Cr_3(D) - Cr_3(D')}{Cr_3(D')} > \epsilon$ (i.e., the percentage of gain in the modularity criteria of average in $D$ is significatively better than $D'$) or

- $\frac{|Cr_3(D) - Cr_3(D')|}{min\{Cr_3(D), Cr_3(D')\}} \leq \epsilon$ and $\frac{SCr_3(D') - SCr_3(D)}{SCr_3(D)} > \epsilon$ (i.e., if both hierarchical partitions are similar in terms of average modularity then the homogeneity criteria has to be significatively better).

In Tables 6–9 we show the comparative analysis in the criteria $(Cr_i, SCr_i)$ for $i = 1, 2, 3$ with $\epsilon = 0.04$ as we have described above (a very similar analysis can be done for any $\epsilon \in [0.01, 0.1]$). For $\epsilon$ values lower than 0.01 the comparison is similar to only consider the modularity criteria and for values greater than 0.1 many dendrograms are considered equal since the difference (from a modularity or homogeneity) is not significative.

**Table 6.** Comparison between GICE and GICEF algorithms versus GN, CNM, Walktrap (Walk), N2012 and Radicchi (Ra) algorithms from a Modularity and Homogeneity point of view in the criteria $Cr_1, \ldots Cr_3$ and $SCr_1, \ldots SCr_3$ for the karate club network. Letter $B$ represent that our algorithm present better performance in this criteria, $W$ worst and $E$ represent that the difference is not significative.

| GICE versus Criteria | GN SC | GN C | GN Both | CNM SC | CNM C | CNM Both | Walk SC | Walk C | Walk Both | N 2012 SC | N2012 C | N2012 Both | Ra SC | Ra C | Ra Both |
|---|---|---|---|---|---|---|---|---|---|---|---|---|---|---|---|
| $(SCr_1, Cr_1)$ | B | E | **B** | W | B | **B** | W | B | **B** | W | B | **B** | B | B | **B** |
| $(SCr_2, Cr_2)$ | B | B | **B** | W | B | **B** | B | B | **B** | W | B | **B** | B | B | **B** |
| $(SCr_3, Cr_3)$ | B | B | **B** | W | B | **B** | B | B | **B** | W | B | **B** | B | B | **B** |
| **GICEF versus Criteria** | **GN SC** | **GN C** | **GN Both** | **CNM SC** | **CNM C** | **CNM Both** | **Walk SC** | **Walk C** | **Walk Both** | **N2012 SC** | **N2012 C** | **N2012 Both** | **Ra SC** | **Ra C** | **Ra Both** |
| $(SCr_1, Cr_1)$ | B | E | **B** | W | E | **W** | W | B | **B** | W | B | **B** | B | B | **B** |
| $(SCr_2, Cr_2)$ | B | B | **B** | W | W | **W** | B | B | **B** | W | B | **B** | B | B | **B** |
| $(SCr_3, Cr_3)$ | B | E | **B** | W | E | **W** | B | B | **B** | W | E | **W** | B | B | **B** |

**Table 7.** Comparison between GICE and GICEF algorithms versus GN, CNM, Walktrap (Walk), N2012 and Radicchi (Ra) algorithms from a Modularity and Homogeneity point of view in the criteria $Cr_1, Cr_2, Cr_3$ and $SCr_1, SCr_2, SCr_3$ for the dolphins network. Letter $B$ represent that our algorithm present better performance in this criteria, $W$ worst and $E$ represent that the difference is not significative.

| GICE versus Criteria | GN SC | GN C | GN Both | CNM SC | CNM C | CNM Both | Walk SC | Walk C | Walk Both | N2012 SC | N2012 C | N2012 Both | Ra SC | Ra C | Ra Both |
|---|---|---|---|---|---|---|---|---|---|---|---|---|---|---|---|
| $(SCr_1, Cr_1)$ | B | W | **W** | B | E | **B** | B | E | **B** | B | B | **B** | B | E | **B** |
| $(SCr_2, Cr_2)$ | B | B | **B** | W | B | **B** | W | B | **B** | B | B | **B** | B | B | **B** |
| $(SCr_3, Cr_3)$ | B | B | **B** | B | E | **B** | B | B | **B** | B | B | **B** | B | B | **B** |
| **GICEF versus Criteria** | **GN SC** | **GN C** | **GN Both** | **CNM SC** | **CNM C** | **CNM Both** | **Walk SC** | **Walk C** | **Walk Both** | **N2012 SC** | **N2012 C** | **N2012 Both** | **Ra SC** | **Ra C** | **Ra Both** |
| $(SCr_1, Cr_1)$ | B | W | **W** | B | W | **W** | B | E | **B** | B | E | **B** | B | W | **W** |
| $(SCr_2, Cr_2)$ | B | B | **B** | W | B | **B** | W | B | **B** | B | B | **B** | B | B | **B** |
| $(SCr_3, Cr_3)$ | B | E | **B** | B | E | **B** | B | E | **B** | B | B | **B** | B | E | **B** |

**Table 8.** Comparison between GICE and GICEF algorithms versus GN, CNM, Walktrap (Walk), N2012 and Radicchi (Ra) algorithms from a Modularity and Homogeneity point of view in the criteria $Cr_1, Cr_2, Cr_3$ and $SCr_1, SCr_2, SCr_3$ for *Les Miserables* network. Letter $B$ represent that our algorithm present better performance in this criteria, $W$ worst and $E$ represent that the difference is not significative.

| GICE versus Criteria | GN SC | GN C | GN Both | CNM SC | CNM C | CNM Both | Walk SC | Walk C | Walk Both | N2012 SC | N2012 C | N2012 Both | Ra SC | Ra C | Ra Both |
|---|---|---|---|---|---|---|---|---|---|---|---|---|---|---|---|
| $(SCr_1,Cr_1)$ | B | E | **B** | B | B | **B** | B | B | **B** | B | B | **B** | B | B | **B** |
| $(SCr_2,Cr_2)$ | B | B | **B** | B | B | **B** | B | B | **B** | B | B | **B** | W | W | **W** |
| $(SCr_3,Cr_3)$ | B | B | **B** | B | B | **B** | B | B | **B** | B | B | **B** | B | E | **B** |
| **GICEF versus Criteria** | **GN SC** | **GN C** | **GN Both** | **CNM SC** | **CNM C** | **CNM Both** | **Walk SC** | **Walk C** | **Walk Both** | **N2012 SC** | **N2012 C** | **N2012 Both** | **Ra SC** | **Ra C** | **Ra Both** |
| $(SCr_1,Cr_1)$ | B | E | **B** | B | B | **B** | B | E | **B** | B | B | **B** | B | B | **B** |
| $(SCr_2,Cr_2)$ | B | B | **B** | W | W | **W** | B | B | **B** | B | B | **B** | B | W | **W** |
| $(SCr_3,Cr_3)$ | B | B | **B** | B | B | **B** | B | B | **B** | B | B | **B** | B | E | **B** |

**Table 9.** Comparison between GICE and GICEF algorithms versus GN, CNM, N2012 and Radicchi algorithms from a Modularity and Homogeneity point of view in the criteria $Cr_1, Cr_2, Cr_3$ and $SCr_1, SCr_2, SCr_3$ for the authors network. Letter $B$ represent that our algorithm present better performance in this criteria, $W$ worst and $E$ represent that the difference is not significant.

| GICE versus Criteria | GN SC | GN C | GN Both | CNM SC | CNM C | CNM Both | N2012 SC | N2012 C | N2012 Both | Radicchi SC | Radicchi C | Radicchi Both |
|---|---|---|---|---|---|---|---|---|---|---|---|---|
| $(SCr_1,Cr_1)$ | B | B | **B** | B | B | **B** | B | B | **B** | B | E | **B** |
| $(SCr_2,Cr_2)$ | B | B | **B** | W | W | **W** | B | B | **B** | B | B | **B** |
| $(SCr_3,Cr_3)$ | B | B | **B** | W | E | **W** | B | B | **B** | E | W | **W** |
| **GICEF versus Criteria** | **GN SC** | **GN C** | **GN Both** | **CNM SC** | **CNM C** | **CNM Both** | **N2012 SC** | **N2012 C** | **N2012 Both** | **Radicchi SC** | **Radicchi C** | **Radicchi Both** |
| $(SCr_1,Cr_1)$ | B | B | **B** | B | E | **B** | B | B | **B** | B | W | **W** |
| $(SCr_2,Cr_2)$ | B | B | **B** | W | B | **B** | B | B | **B** | B | B | **B** |
| $(SCr_3,Cr_3)$ | B | B | **B** | W | B | **B** | B | B | **B** | E | E | **E** |

## 7. Computational Results in Random Networks

Following the Girvan-Newman benchmark defined in [4], we have generated random graphs with 128 vertices and 4 communities $C_1, \ldots, C_4$ of size $|C_i| = 32$ each. Every vertex has an expected degree of $< k >= 16$. Regarding the expected out-degree, which quantifies the expected amount of neighbours of a node which is assigned to a different cluster and is denoted by $z_{out}$, takes values from 1 to 10. Then, it is clear that the higher values of out-degree are related to networks whose community structure is quite weak. Our aim with this experiment is to measure or somehow quantify the sensitivity of the methods compared, by considering the strength of the communities obtained. On the other hand, $z_{in}$ denotes the expected number of neighbors of a node which is assigned to the considered community $C_i$. In order to establish if an edge is included or not in a cluster, we generate a random synthetic graph according to the probability distribution function showed below. Then, for every pair of nodes $i, j$. the probability that there exists the edge $\{i, j\}$ is calculated by:

$$P(i,j) = \begin{cases} \alpha & if \ \text{i and j belong to } C_k \\ \beta & \text{otherwise} \end{cases}$$

From the values of $z_{out}$ and $z_{in}$ it is calculated the value $\mu = \frac{z_{out}}{z_{in}+z_{out}}$, which is called mixing parameter. Let us observe that for any node $i$, $< k_i >= 16 = z_{in} + z_{out} = (31)\alpha + (96)\beta$. So from the value of $\mu$ or $Z_{out}$, it can be generated the graph in this GN framework. Generally speaking, almost all the methods fail when the value of $\mu$ is greater than 0.4, as it happens with *GN*, *Radicchi*, *Walktrap* or *CNM* [22]. However, they work properly when $\mu \leq 0.4$. In [22], it can be found with more details the results given by the *GN* and *CNM*. In Table 10 we show the results obtained with the *GICE* algorithm, and *GICEF* algorithm for the step 4. In order to measure the similarity between the reference partition $P = \{C_1, \ldots, C_4\}$ with $C_1 = \{1, \ldots, 32\}, C_2 = \{33, \ldots, 64\}, C_3 = \{65, \ldots, 96\}$ and $C_4 = \{97, \ldots, 128\}$ we have used the *Normalized Mutual Information* (NMI). In the table we show, for each scenario, the average over 100 iterations realized over the benchmark graph. In this table we also show the modularity average in the 100 realization of the *GICE*, *GICEF* and also the modularity average of the reference partition.

**Table 10.** Normalized Mutual Information (NMI) based on the mixing parameter $\mu = \frac{z_{out}}{z_{in}+z_{out}}$ respect to the reference partition with four groups in the *GN benchmark* using the *GICE* and *GICEF* algorithms.

| $\mu$ | $z_{out}$ | $< k >$ | *Reference* Modularity | GICE NMI | GICE Modularity | GICEF NMI | GICEF Modularity |
|---|---|---|---|---|---|---|---|
| 0.1 | 1.6 | 16 | 0.6504 | 1 | 0.6504 | 1 | 0.6504 |
| 0.2 | 3.2 | 16 | 0.5502 | 1 | 0.5502 | 1 | 0.5502 |
| 0.3 | 4.8 | 16 | 0.4491 | 0.9932 | 0.4492 | 0.9660 | 0.4491 |
| 0.35 | 5.6 | 16 | 0.4000 | 0.9593 | 0.3972 | 0.9290 | 0.3927 |
| 0.40 | 6.4 | 16 | 0.3507 | 0.8925 | 0.3429 | 0.8260 | 0.332 |
| 0.45 | 7.2 | 16 | 0.3002 | 0.7914 | 0.2885 | 0.6605 | 0.2624 |
| 0.5 | 8.0 | 16 | 0.2504 | 0.5500 | 0.2361 | 0.3979 | 0.1931 |
| 0.55 | 8.8 | 16 | 0.1999 | 0.2979 | 0.1959 | 0.2078 | 0.1657 |
| 0.6 | 9.6 | 16 | 0.1502 | 0.1351 | 0.1901 | 0.1010 | 0.1573 |

## 8. Discussion

In this section we discuss the results obtained in the different tests. Regarding the well-known examples, in Tables 2–5 we show the results in terms of modularity and homogeneity of the five hierarchical clustering algorithms previously mentioned for the first partitions in those examples. The criteria $Cr_1, Cr_2, Cr_3, SCr_1, SCr_2$ and $SCr_3$ are also analyzed. We can say that, in general, the *GICE* algorithm here presented and its fast version are very competitive from modularity and homogeneity point of view.

Moreover, let us note that the fast version of the algorithm here proposed presents worst performance than the slower version, since the fast version obtain results significatively worst in 9 of 12 scenarios. Nevertheless, comparing this fast version with the rest of the algorithms studied in this paper, we see that it is very competitive. The fast version of the algorithm here presented improves the results of the *CNM* (see Tables 6–9) in 7 of 12 scenarios, improves the results of Radicchi algorithm in 8 of 12 (only is worst in 3 of 12) situations, present better results than *GN* and *N*2012 algorithms in 11 of 12 scenarios and improves the results of *Walktrap* in all situations.

Regarding the performance of our algorithm in the benchmark model, several conclusions previously explained are repeated again. As happen with the well-known examples, it is clear that the performance of the *GICE* is better than the fast version. Both provide very good results and similar *NMI* values for $\mu$ lower than 0.3, but for $\mu = 0.35$ the difference is 0.03, for $\mu = 0.4$ the difference is 0.07, and for $\mu = 0.5$ the difference is 0.15, so the difference between these two algorithms increased (in terms *NMI*) as the value of $\mu$ increase. Nevertheless, note that the performance of *GICEF* is better than other fast algorithms compared in this paper as the CNM for these values. For example, (see [22]) *NMI* of *CNM* for $\mu = 0.3$ is around 0.9, for $\mu = 0.4$ is around 0.75, and for $\mu = 0.5$ is around 0.38. In general, *GN*, Walktrap, *N*2012 and *Radicchi* algorithms give worst results than *CNM*.

To conclude this section, let us note that in general and for $\mu$ values lower than 0.55, the reference partition is the best partition in the criteria $(SCr_1, Cr_1)$, since this partition has the best possible coefficient of variation (its homogeneity is perfect) and its modularity is better or at least non significantly lower (for values of $\mu = 0.45, 0.5, 0.55$ it can be found partitions with higher values of modularity but not significatly) than other partitions founded by *GICE* algorithm. Taking into account this, *NMI* could reflect in some way both criteria $(SCr_1, Cr_1)$ (as we have done in the previous section) since it is clear that if one partition is more similar to the reference one than other, then it is expected better values of $(SCr_1, Cr_1)$. A similar analysis, could be done taking into account (for the same random graphs) what happen in the second and the third step to consider the criteria $(SCr_2, Cr_2)$ and $(SCr_3, Cr_3)$. Also let us note that for $\mu$ values greater than 0.60, the *GICE* and *GICEF* algorithms improve the reference modularity.

## 9. Final Remarks and Conclusions

Community detection problems [1] are commonly described as clustering network problems whose main aim is to obtain a good partition of the graph analyzed. In this paper we formally present the *hierarchical clustering network problem* (HCNP) as the problem to find a *good* hierarchical partition of a network. This new perspective emphasizes more in the dynamic process of the clustering rather than the final *picture* of the clustering process. From a social network analysis, a hierarchical partition of a network is more informative than a partition, as it shows the whole process along which the communities are grouped (regarding agglomerative methods) or splitted (regarding divisive methods), from the very first step until the last situation. As Clauset et al. point out in [46], *a hierarchy is a central organizing principle of complex networks, capable of offering insight into many network phenomena.* Hence, it can be said that a hierarchical analysis of a graph provides much more information than the output of a single non-hierarchical method.

Taking into account the HCNP approach here introduced, in this paper we propose a divisive algorithm based on a new SP edge betwenness measure. Given a network for which the only information is the direct relations among its members, a common assumption for calculating indexes such as the SP edge betweenness measure is that the communication between any pair of nodes in the network is equally important. The key element of this study is to avoid this assumption, and then to estimate the importance of communications between any pair of nodes (not necessarily adjacent) in a network. Then, we propose a new SP betweenness measure in which the communications between nodes are not equally weighted. To measure the importance of the communication between

two nodes $i$ and $j$ (i.e., $w_{ij}$), we first measure the importance or power of these nodes, assuming that the communications between important nodes is more relevant that the communication between dummy nodes. The power of nodes in network is obtained as the Shapley value (or any other semivalue) of a game that we call linear modularity game. Let us emphasize again that this game has been built under the assumption that the only available information in the problem is the network, so only modularity function has been considered. If more information is known about the clustering problem, this information should be included in the game, to obtain more cohesive partitions [47,48].

To measure the performance of the divisive hierarchical clustering network algorithm here proposed, we agree on the need for having different methods to evaluate the performance of community detection and classification algorithms, whenever the data shows some structure [46]. In the framework of algorithms for the clustering of networks, we can find [1] some measures to assess the quality of any particular partition of a graph. For example, in [4] it is introduced the notion of modularity; we also considered the homogeneity size to measure the quality of a partition. However, these measures are not well fitted to the performance assessment of a hierarchical algorithm, in which the final output is usually a dendrogram. Hence, in this paper we introduce several criteria that allow us to compare different dendrograms of network from a modularity and homogeneity point of view. Nevertheless, more efforts should be dedicated in community detection literature to compare two hierarchical partitions of a given graph.

The computational results reveal that the hierarchical clustering network algorithm here defined eliminates the main problems of the classical $GN$ algorithm for many real situations and exhibits good performance in those situations in which the classical betweenness measure also does. We observe that partitions obtained using the our $GICE$ algorithm (in which the node-game SP edge betweennees measure is used) yield groups that are more homogeneous in size than the classical $GN$ algorithm (in which the classical SP edge betweenees is used).

Another crucial concept we develop is the use of the minimum operator to aggregate the importance of two nodes to represent the importance of the corresponding communication. Other conjunction aggregation operators could also be used instead. The relations between the results for a divisive algorithm and the aggregation operators is an issue that merits investigation.

Most of social network analysis, as centrality measures or community detection problems among others, analyze the network and its properties only taking into account the relations that exist between nodes (i.e., the graph) or in more complex situations assuming that these relation are valued (in a positive or negative way). Nevertheless, as it was pointed in [16,34], it is very difficult to find a social network analysis in which it is also reflected the interests or motivations that reflect the interaction between nodes in a network. In [34], it was introduced a game theory framework to model (in addition with the relation that exist between nodes in a network) the interactions that motivates the relation between nodes in the network by means of a cooperative game. From a community detection point of view, let us suppose that we have a wheel-graph with four nodes $N = \{1, 2, 3, 4\}$ ordered in a natural way (i.e., $E = \{(1, 2), (2, 3), (3, 4), (1, 4)\}$ ) that we want to cluster. If we have not additional information about what is the real situation that are represented by this graph or we do not know the interests that have motivated this network to be built, then the first partition could be $P = \{1, 2\}; \{3, 4\}$ or $P' = \{1, 4\}; \{2, 3\}$. Now, let us suppose that the interest which motivates the relation between nodes in this network is to maintain together the nodes $1 - 4$ and $2 - 3$. Then, the characteristic function $v(S)$ that models the worth associated with the coalition $S$, can be $v(\{i\}) = 0$, for all $i = 1, \ldots, 4$, $v(\{1, 2\}) = v(\{1, 3\}) = v(\{2, 4\}) = v(\{3, 4\}) = 0$, $v(\{1, 4\}) = v(\{2, 3\}) = 1$, $v(S) = 1$ if $|S| = 3$ and $v(\{1, 2, 3, 4\}) = 2$ since in this coalition we maintain $1 - 4$ and $2 - 3$ together. From a clustering network point of view, it is clear that if the interest of this network is to maintain the nodes $1 - 4$ together or $2 - 3$, then the partition $P = \{1, 2\}; \{3, 4\}$ is not realistic for this situation. We want to emphasize that the divide algorithm here defined

could take into account the interests or motivations that have the nodes in the network by means of a cooperative game and could take into account this information in the clustering process of the network. Actually, there is not any hierarchical or non hierarchical clustering network algorithm that take into account this information for the clustering problem.

To conclude this section, we want to emphasize that the divisive algorithm here presented based on the node-game SP betweenness measure clearly provides very good results in terms of modularity and homogeneity. For small and medium networks, its complexity (the same that the GN algorithm) allows us to obtain a very informative evolution of how the groups are break in the network in a hierarchical way, and it should be the best option to analyze these size networks from this point of view. For large networks, we present a simplification of our measure, providing a fast version of the algorithm that has a quadratic order on sparse networks. This fast version is competitive from a computational point of view with other hierarchical algorithms (as for example *CNM*, *Walktrap*, *N*2012, *GN* or *Radicchi*) presenting (in general) better results in the examples analyzed.

**Author Contributions:** Conceptualization, J.C. and D.G.; methodology, J.C. and D.G.; software, J.C. and I.G.; validation, D.G. and R.E.; formal analysis, J.C. and D.G.; investigation, J.C. and D.G.; resources, I.G. and R.E.; data curation, I.G. and R.E.; writing—original draft preparation, D.G., J.C. and I.G.; writing—review and editing, I.G. and R.E.; visualization, I.G.; supervision, D.G., J.C. and R.E.; project administration, I.G.; funding acquisition, D.G., J.C. and R.E. All authors have read and agreed to the published version of the manuscript.

**Funding:** This research was funded by the Government of Spain, Grant Plan Nacional de I+D+i [PID2020-116884GB-I00 and PGC2018096509-B-I00] and the Complutense University of Madrid [PR108/20-28 and CT17/17-CT18/17.

**Institutional Review Board Statement:** Not applicable.

**Informed Consent Statement:** Not applicable.

**Data Availability Statement:** Not applicable.

**Conflicts of Interest:** The authors declare no conflict of interest.

## Appendix A

In this Appendix we demonstrate that Equation (6) and (7) are equivalent.

By definition, $l_{in}(C_r) = \dfrac{\sum_{i,j \in C_r} A(i,j)}{2}$, and $l_{out} = \sum_{\substack{i \in C_r \\ j \notin C_r}} A(i,j)$. Also, we know that:

1. $\displaystyle \sum_{i,j \in C_r} l_i l_j = l_{i_1}^2 + l_{i_2}^2 + \cdots + l_{i_{n_r}}^2 + 2 l_{i_1} l_{i_2} + \cdots + 2 l_{i_{n_r-1}} l_{i_{n_r}} = \left( \sum_{i \in C_r} l_i \right)^2$, being $n_r$ the number of elements in the cluster $C_r$.

2. $\dfrac{l_{in}(C_r)}{m} + \dfrac{l_{out}(C_r)}{2m} = \dfrac{\sum_{i,j \in C_r} \frac{A(i,j)}{2}}{m} + \dfrac{\sum_{\substack{i \in C_r \\ j \notin C_r}} A(i,j)}{2m} = \dfrac{\sum_{i \in C_r, \forall j} A(i,j)}{2m} = \dfrac{\sum_{i \in C_r} l_i}{2m}$

Then, it holds that

$$\frac{1}{2m} \sum_{r=1}^{k} \sum_{i,j \in C_r} \left[ A(i,j) - \frac{l_i l_j}{2m} \right] = \sum_{r=1}^{k} \left[ \frac{1}{2m} \sum_{i,j \in C_r} A(i,j) - \frac{1}{(2m)^2} \sum_{i,j \in C_r} l_i l_j \right] \overset{1.}{=}$$

$$\sum_{r=1}^{k} \left[ \frac{l_{in}(C_r)}{m} - \frac{\left( \sum_{i \in C_r} l_i \right)^2}{(2m)^2} \right] = \sum_{i=1}^{k} \left[ \frac{l_{in}(C_r)}{m} - \left( \frac{\sum_{i \in C_r} l_i}{2m} \right)^2 \right] \overset{2.}{=}$$

$$\sum_{r=1}^{k} \left[ \frac{l_{in}(C_r)}{m} - \left( \frac{l_{in}(C_r)}{m} + \frac{l_{out}(C_r)}{2m} \right)^2 \right].$$

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
