# Peer review of "A New Edge Betweenness Measure Using a Game Theoretical Approach: An Application to Hierarchical Community Detection"

_mathematics, doi:10.3390/math9212666_

Round 1

Reviewer 1 Report

This paper analyzes the custering network problem, with a methodology based on concepts from cooperative game theory.

The presentation is clear and accurate, and a novel algorithm is presented. Overall, the whole exposition is self-consistent.

My impression is that the paper deserves pubblication, and that the only thing that could be improved is the English exposition.

Indeed, sometimes the language is not fluid and could be improved by a mother tongue proof reading.

Author Response

Thank you for all  the comments. We attach a point-by-point response.

Reviewer 2 Report

The article is devoted to hierarchical methods for determining the coalition structure of the network. A hierarchical partition is defined as a sequence of partitions in which the number of clusters increases monotonically by one. At the same time, to determine the important edges that will be removed, it is proposed to determine the betweenness centrality as a weighted sum of the shortest paths passing through this edge. The weights in this sum are suggested to be chosen as the solution of a cooperative game. To do this, they construct a characteristic function in the form (8), which includes the number of edges inside the cluster and leaving the cluster to other clusters. They prove three propositions on the value of such a cooperative game in the form of the Shapley value. In this sense the approach is closed to the paper "Network partitioning algorithms as cooperative games", by Avrachenkov, Kondratev, Mazalov, Rubanov. Computational Social Networks, 5:11. 2018. Pp. 1-28. DOI: 10.1186/s40649-018-0059-5. There the characteristic function is determined as a number of shortest paths in coalition S, and the Shapley-Myerson value is used as a solution of the game. Another approach  from this paper is to use the modularity as an objective function in potential game.  Notice that the characteristic function in this paper is superadditive.

It is not the case in the presented paper. Consider for example the graph N={1,2,3,4,5} with E={(1,2), (1,3), (1,4), (2,3), (2,5), (3,4)}.  If S={1} then v_{mod}(S)=-3\beta/6. If S={1,2} then v_{mod}(S)=(\alpha-4\beta)/6. So, it is not monotone if \alpha<\beta. It is not good to use Shapley value. 

The authors wrote that the function (8) is an analogue of the modularity (7). But the equation in (7) is quadratic and for large C_r the coefficient in l_{in}(C_r)  is negative. So, for huge C_r the edges inside the cluster is also punishment.

 Also, please clarify the proof of the propositions. First, please explain the definition of S^k (line 350). Consider the example 3. What are the S^0, S^1,... and waht is l_i for the node 2 and 4. 

To conclude, I think that the proposed approach is new, but it causes criticism and requires certain explanations. 

Author Response

(The authors gave the same response as above.)

Round 2

Reviewer 2 Report

The authors supplemented the text with several examples, which undoubtedly improved the text of the article. Propositions 1-3 have become clearer. The reviewer has only a few comments left.
1. Prove the transition from formula (6) to formula (7) for the modularity. Give an example.
2. According to the proven propositions, the solution of a cooperative game with a linear characteristic function in the form of semivalues, as well as in the form of the Shapley and Banzaf-Coleman values, depends only on the positive part of the function, i.e. on the number of edges inside the cluster and does not depend on the number of edges coming out of the cluster. Although the proof is carried out correctly, it needs to be explained why this happened.
The results of the paper are new and interesting, but  additions could significantly improve the text for the readers.

Author Response

Thank you for your comments and advice. Here we include our comments about the points mentioned.

Round 3

Reviewer 2 Report

The authors have addressed my previous comments. I am happy with the current version and recommend its acceptance without further changes.